# Auditory deep sleep stimulation in older adults at home: a randomized crossover trial

Caroline Lustenberger [1,2,3 ✉], M. Laura Ferster[4], Stephanie Huwiler [1], Luzius Brogli[4,5], Esther Werth [2,3,6], Reto Huber [2,3,7,8] & Walter Karlen [3,4,5]

## Abstract

**Background** Auditory stimulation has emerged as a promising tool to enhance non-invasively sleep slow waves, deep sleep brain oscillations that are tightly linked to sleep restoration and are diminished with age. While auditory stimulation showed a beneficial effect in lab-based studies, it remains unclear whether this stimulation approach could translate to real-life settings.

**Methods** We present a fully remote, randomized, cross-over trial in healthy adults aged 62–78 years (clinicaltrials.gov: NCT03420677). We assessed slow wave activity as the primary outcome and sleep architecture and daily functions, e.g., vigilance and mood as secondary outcomes, after a two-week mobile auditory slow wave stimulation period and a two-week Sham period, interleaved with a two-week washout period. Participants were randomized in terms of which intervention condition will take place first using a blocked design to guarantee balance. Participants and experimenters performing the assessments were blinded to the condition.

**Results** Out of 33 enrolled and screened participants, we report data of 16 participants that received identical intervention. We demonstrate a robust and significant enhancement of slow wave activity on the group-level based on two different auditory stimulation approaches with minor effects on sleep architecture and daily functions. We further highlight the existence of pronounced inter- and intra-individual differences in the slow wave response to auditory stimulation and establish predictions thereof.

**Conclusions** While slow wave enhancement in healthy older adults is possible in fully remote settings, pronounced inter-individual differences in the response to auditory stimulation exist. Novel personalization solutions are needed to address these differences and our findings will guide future designs to effectively deliver auditory sleep stimulations using wearable technology.

## Plain language summary

Sleep's restorative function is closely linked to slow waves, which are brain activity patterns that occur during deep sleep and are diminished with age. Those slow waves can be increased through auditory stimulation, a method that administers precisely-timed sounds during sleep. Here, we established whether the application of auditory stimulation can be performed in older adults over several nights remotely in their own homes. In a trial, we used a mobile device to deliver auditory stimulation during sleep and measured its effects on slow waves, mood, and vigilance. Although we showed robust increases in slow waves, we found large effect differences between participants and also between different nights within the same participants. We looked for predictors of these effect differences. Our study showed that in-home auditory stimulation is feasible, and may help to guide future auditory stimulation strategies.

[1] Neural Control of Movement Lab, Institute of Human Movement Sciences and Sport, Department of Health Sciences and Technology, ETH Zurich, Zurich, Switzerland. [2] Neuroscience Center Zurich (ZNZ), University of Zurich and ETH Zurich, Zurich, Switzerland. [3] Center of Competence Sleep & Health Zurich, University of Zurich, Zurich, Switzerland. [4] Mobile Health Systems Lab, Department of Health Sciences and Technology, ETH Zurich, Zurich, Switzerland. [5] Institute of Biomedical Engineering, Universität Ulm, Ulm, Germany. [6] Department of Neurology, University Hospital Zurich, University of Zurich, Zurich, Switzerland. [7] Child Development Centre, University Children's Hospital, University of Zurich, Zurich, Switzerland. [8] Department of Child and Adolescent Psychiatry and Psychotherapy, Psychiatric Hospital Zurich, University of Zurich, Zurich, Switzerland. ✉email: caroline.lustenberger@hest.ethz.ch

While basic and applied research in promoting human health has increased substantially in the last decades, its translation into clinical applications with access for a wider public has lagged behind[1–3]. In contrast, there is an ever-growing range of mobile health devices and technologies that found their way into the consumer market and into people's homes, yet clinical validations of most of these applications is lacking[4,5]. With these technologies, an important platform has been developed to move the findings from well-controlled lab studies to in-field applications[6]. However, to establish the potential and limitations of preventive and therapeutic health approaches in real-life settings, studies are needed that identify individual differences, establish predictors for successful applications, and elucidate intended and unintended effects of the prolonged intervention in different contexts.

In this regard, translating in-lab methods that promote healthy sleep have gained increased attention because sleep could be a key player in promoting brain and body health up until old age[7,8]. Sleep in older adults is hallmarked by reduced sleep duration, increased sleep fragmentation, and reduced sleep efficiency[9–11]. Most pronouncedly, sleep becomes more superficial with age, which is reflected in a decrease of sleep slow waves[12,13]. Slow waves are dominant low-frequency (i.e., <4 Hz) brain rhythms during deep non-rapid eye movement (NREM) sleep, which are considered to mirror sleep intensity and are fundamental for healthy and consolidated sleep[14]. They have been implicated in promoting brain plasticity[15], memory formation[16], immune-supportive functions[17], and have recently been inversely related to amyloid burden in older adults[18,19]. Therefore, deep sleep enhancement strategies will potentially contribute to healthy aging, which is highly relevant considering the growing proportion of older people in our society. Specifically, the enhancement of slow waves might be an ideal target to improve sleep quality. Consequently, applications that enhance slow waves in a non-pharmacological, non-invasive, and physiological way have become a central topic in sleep research.

Among others, the application of acoustic stimuli during sleep have crystallized as a very promising avenue. In 2013, Ngo and colleagues[20] reported that acoustic stimuli specifically targeted to the up-phase of the endogenous slow wave can enhance slow wave amplitude along with improved declarative memory in young participants. Thereafter, several in-lab studies from different research groups replicated the beneficial effect of phase-targeted or rhythmic (i.e., around 1 Hz) acoustic stimuli to boost slow waves mainly in young participants, but also in middle-aged, and older populations[20–30]. These slow wave enhancement effects particularly benefited declarative memory consolidation[20–25,31], but single studies have also reported benefits for executive[29], immune-supportive[17], and autonomic functions[28]. Collectively, there is convincing evidence that slow waves can be enhanced in in-lab settings during single session applications along with some specific brain and body functions. Owing to these first in-lab results, this deep sleep modulation approach has already found its way into the consumer-market[32,33]. However, no auditory sleep stimulation technology has yet established itself as certified medical wearable. To date, no controlled, randomized clinical trial exists that investigated the effect of auditory up-phase stimulation over multiple nights, in at-home settings and its effects on sleep and daytime functioning.

In order to overcome this gap, we recently developed a portable sleep monitoring and feedback-controlled slow wave modulation device with research-grade accuracy[34]. Here, we used this system in the first randomized controlled cross-over clinical trial applying auditory stimulation during slow wave sleep over multiple weeks in healthy older adults between 60 and 80 years of age and without diagnosed sleep disorders in order to establish effectiveness thereof in enhancing slow wave activity as our primary outcome. Older adults are of specific interest because of their reduced slow waves and their less pronounced potential to benefit from auditory stimulation as revealed in an in-lab study[35]. The device was worn at home and self-applied by the participants. Their sleep schedule was not restricted to a specific schedule. In addition, we implemented mobile phone based daily ecological momentary assessments (EMA) of mood, subjective sleep quality, and vigilance to determine the direct effects of stimulation on these secondary outcome parameters. We demonstrate on the group level that slow waves can be enhanced over multiple days in older adults, also in non-controlled, in-home settings, and for two different stimulation protocols. We establish predictors that define participants with strong and weak responses to auditory stimulation, and predictors that relate to nightly variance within participants. While slow wave enhancement in healthy older adults is possible in at-home settings and could potentially benefit related functions, we must consider that clear inter-individual differences in the response to auditory stimulation exists and novel personalization solutions are needed to address these differences to achieve high efficacy and effectiveness in the field.

## Methods

**Participants**. Overall, 33 participants were enrolled and screened (first participant enrolled: 07.05.2018, last participant out: 05.03.2019, study has been completed). An overview of the trial profile of the enrolled participants is shown in Supplementary Fig. S1. Here, we report data of 16 participants that underwent the whole experimental procedure of auditory up-phase slow wave stimulation over multiple weeks and had sufficient data for analysis. This dataset was analyzed as part of the planned interim analysis of this clinical trial (NCT03420677), however, also represents the final results because the study was terminated early due to the COVID-19 pandemic. The included participants (nine males, seven females, age [mean ± SEM (standard error of the mean)]: 69.5 ± 1.3 years, age range: 62–78 years) were all in good general health, non-smokers and had a stable home situation that allowed for reliable application of intervention for the duration of the study. None of them had a presence or history of a psychiatric/neurologic disorder, presence of a diagnosed sleep disorder or an internal disorder. Participants were further excluded if they used on-label sleep medication, showed known or suspected non-compliance, drug or alcohol abuse, cannabis or nicotine intake, or excessive caffeine consumption. We further confirmed with basic audiometry that they were able to hear at least the intervention sound at 50 dB on both ears (hearing threshold [mean ± SEM]: 44.1 ± 0.8 dB, range: 40–50 dB). The participants were recruited from the community using advertisements on different platforms/newspapers and at meet-ups. The recruitment was further supported by the University Zurich Research Priority Program "Dynamics of Healthy Aging" and the Senior University. All participants were compensated for their participation. The study was approved by the cantonal Ethics Committee Zurich (reference number: KEK ZH, BASEC 2017-01436), the Swiss Agency for Therapeutic Products Swissmedic (reference number: 10000181/2017-MD-0027) and was conducted in accordance with the Declaration of Helsinki. The study was registered at ClinicalTrials.gov (NCT03420677). All the participants provided written informed consent before participation. Participants completing the full study duration received at least 400 Swiss Francs for participation. In addition, up to 200 Swiss Francs could be earned proportionally to the completed tasks during the study. The identifiable person in Fig. 1a, who was not a participant in this study, provided written consent to publish this image.

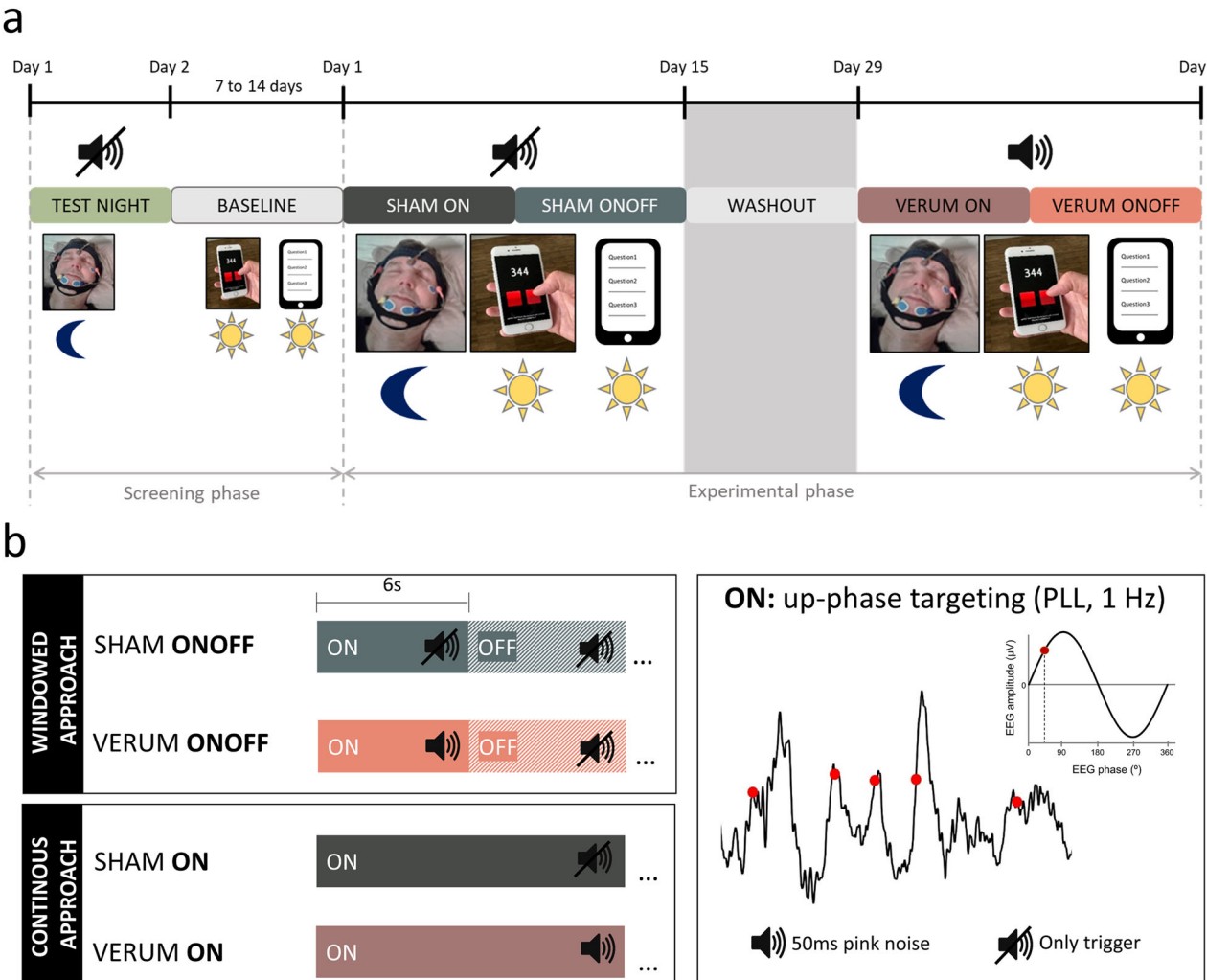

**Fig. 1 Study and auditory stimulation design. a** Participants first underwent a screening phase to familiarize with the in-home MHSL-Sleepbandv2 (MHSL-SB) use and the mobile phone assessments. In the experimental phase, two-week intervention periods of either Sham (no auditory stimulation) or Verum (auditory stimulation) were conducted in a randomized, crossover design. Daytime recordings reported here (hallmarked by a sun) included subjective sleep quality assessment in the morning, a psychomotor vigilance test (PVT) at 2 pm, the Karolinska sleepiness scale (KSS) in the morning, at 2 pm, and 8 pm, and a mood assessment in the morning and at 8 pm. Nocturnal recordings (hallmarked by a moon) included the nightly application of the MHSL-SB that recorded Fpz-A2 (electroencephalography [EEG]), left and right electrooculography (EOG), and chin electromyography (EMG). The identifiable person in this figure provided written consent to publish this image. **b** The MHSL-SB performed a sleep classification and defined stable NREM sleep and used a phase-locked-loop (PLL) to precisely time the slow wave modulation during up-phase of the slow wave. During Sham, only triggers were saved to mark slow wave up-phases, during Verum 50 ms pink noise bursts were applied (illustrated as red dots). The defined PLL target phase was around 45°. In each intervention period two types of application were used, the continuous and windowed approach. In the windowed approach, 6 s ON and 6 s OFF windows were alternated, during OFF no slow waves were targeted (but marked), during ON slow wave up-phases were targeted and either stimulated with pink noise bursts (Verum) or only marked (Sham). During the continuous approach, all targeted slow wave up-phases were either stimulated (Verum) or marked (Sham). Each of these approaches lasted for 7 days in a counter-balanced design resulting in four possible order combinations that were counter-balanced across participants: (1) Verum ON, Verum ONOFF, Sham ON, Sham ONOFF, ($n_{subjects}$ = 4) (2) Sham ON, Sham ONOFF, Verum ON, Verum ONOFF, ($n_{subjects}$ = 3) (3) Verum ONOFF, Verum ON, Sham ONOFF, Sham ON, ($n_{subjects}$ = 5) (4) Sham ONOFF, Sham ON, Verum ONOFF, Verum ON ($n_{subjects}$ = 4).

**Experimental procedure**. The experimental procedure is summarized in Fig. 1. Participants completed a six-week randomized, crossover experimental protocol in which they had two weeks of auditory slow wave stimulation (Verum) and two weeks of Sham interleaved with a two-week washout period (Fig. 1a). A crossover design was chosen because the within-participant variation of sleep is less than the between-participant variation and therefore, fewer participants are required for this study compared to a randomized parallel-group design. Possible disadvantages of the crossover design such as larger dropout rate, carry-over effects, or instability of the participant's condition were not expected in

this study. The study was conducted in Switzerland at the participant's home. All participants first underwent a screening phase and during an initial home visit several demographic questionnaires were answered. They were further extensively instructed on how to use the intervention device and the EMA on the mobile phone. We also performed a basic audiometry to assure that their hearing ability was good enough to perceive the sound level of the intervention device on both ears. The first night habituated the participant to the MHSL-SleepBandv2 (MHSL-SB, ETH Zurich, Zurich, Switzerland) and used EMA. If participants fulfilled all criteria and agreed to participate in the experimental

sessions, a baseline period followed. Thereafter, the experimental phase started, including two 2-week intervention periods that were interleaved with a 2-week washout period. One intervention period was including auditory stimulation of slow wave sleep (Verum), the other one was not including auditory stimulation (Sham). The intervention periods were further subdivided into two different stimulation approaches (windowed and continuous, counter-balanced order) that are summarized in Fig. 1 and will be explained in the paragraph MHSL-SB and auditory stimulation. The condition applied was in a counter-balanced, randomized, and double-blind design. Randomization was performed by the study coordinator that was not involved in recruitment or data collection but only analysis. Participants were randomized in terms of which intervention condition will take place first, having four different options (Verum ON, Verum ONOFF, Sham ON, Sham ONOFF) using a blocked design to guarantee balance. After the first condition was defined, the following conditions were set (Sham always followed Sham, and ON/ONOFF were kept in the same order, the same was for Verum). Gender was taken into account when performing randomization in order to achieve an equal gender balance. The randomization order was predefined in blocks for both genders before the study started and then participants were randomly allocated by the study coordinator to the first condition in sequence after their successful enrollment upon passing screening performed by trained and blinded research assistants. The randomization aimed at 25% ($n = 4$) start per condition, however, due to intermediate drop-outs (see data exclusion above) and gender imbalance, this counter-balance was not perfectly achieved but very close to the aimed distribution: Start Sham ONOFF: 4 (2f, 2m), Sham ON: 3 (1f, 2m), Verum ONOFF: 5 (2f, 3m), Verum ON: 4 (2f, 2f). Each start, the middle, and end of an intervention period was accompanied by home visits during which experimenters provided or collected the MHSL-SB, respectively. The participants and experimenters performing the assessments and providing the device/instructions were blinded to the conditions. The study coordinator, PI, and lead engineer had to know conditions for device settings and safety reasons. Experimenters performing the assessments were blinded to the performed intervention by restricting access to device settings only to the lead engineer and randomization list access only to the study coordinator. Participants were blinded by having for both Sham and Verum conditions exactly the same application procedures of the device and the exact same physical appearance of the device independent of intervention setting. During the complete intervention period of 14 nights, participants were wearing the MHSL-SB during the night to collect EEG, EOG, and EMG data and to perform auditory stimulation in case of the Verum condition. The next day they performed EMAs in the morning, at 2 pm and at 8 pm (see paragraph *Daytime assessments with Mobile Phone Apps*). Collectively, data reported in this manuscript include the recordings of the MHSL-SB during the intervention period and the EMA collected in the morning (sleep quality), 2 pm and 8 pm, which represent our secondary outcome variables. Additional assessments of the quality of life questionnaires and cognitive tasks were performed before and after each intervention period of the study that will not be reported here. The analysis of these questionnaires and cognitive outcomes are underpowered due to the early termination because of the COVID-19 pandemic to draw meaningful conclusions from these parameters. Furthermore, cardiovascular assessments were performed daily but will not be reported here.

**MHSL-SB and auditory stimulation.** Biosignals were recorded using the MHSL-SB, a mobile and configurable system for real-time sleep recording and processing which also implemented the auditory slow wave stimulation[34]. One single EEG signal (Fpz-A2) was used for real-time processing, while two EOG and two EMG signals were recorded for post-processing purposes. Single-use auto-adhesive electrodes (Neuroline 720, Ambu A/S, DK) enabled self-application while keeping signal quality high. Biosignals were sampled at 250 Hz using a 24-bit analog-to-digital converter that featured an on-chip anti-aliasing filter. The raw biosignals including AC and DC components were saved to an SD card and further piped to the embedded real-time algorithms for auditory stimulation. To produce precise auditory stimulation, the MHSL-SB implemented a set of algorithms. The single EEG signal was processed to eliminate power line noise (first-order notch filter, 50 Hz). Then, overall, a stimulation decision logic triggered the auditory stimuli when NREM sleep, slow wave sleep (SWS), low beta power, and EEG target phase conditions were simultaneously met. First, the NREM sleep algorithm categorized the EEG data as NREM sleep or not-NREM sleep (awake and REM) based on the spectral power at different frequency bands: low delta ([2–4] Hz), high delta ([3–5] Hz), and high beta ([20–30] Hz). If low and high delta powers were above, and beta power below predefined and fixed thresholds during a specific window length, the EEG signal was classified as NREM sleep. When the EEG data was classified as NREM sleep for the first time, this algorithm immediately enabled the follow-up algorithms. However, stimulation triggers were enabled only after 10 min of consecutive NREM sleep is detected to ensure stable NREM sleep. This 10 min period was also used as control baseline. For the follow-up NREM sleep classification periods the stable NREM sleep time was decreased to 3 min. Metrics regarding algorithm performance are summarized in Supplementary Table S1. Algorithm performance was quantified using precision, specificity, and recall. Precision indicated the rate of correctly predicted NREM sleep samples. Specificity indicated the ability of the sleep classification algorithms to identify not-NREM sleep correctly. Finally, the recall metric indicated the correct NREM sleep classification rate when the data was also scored as NREM sleep. All metrics were defined as relative numbers ranging from 0 to 1.

To increase stimulation accuracy during deep NREM sleep, a SWS classification and beta power detection algorithm were implemented. When delta power overpassed a predefined threshold condition, the EEG signal was classified as SWS. Since power increment in the beta frequency range has been related to awake, light sleep, artifacts, and the presence of arousals, a beta power detection algorithm prevented stimulations when beta power was exceeding a pre-defined threshold in the last second.

The algorithm used to deliver the auditory stimulation in phase with the SW consisted of a first-order phase-locked loop (PLL) that estimated the EEG phase in real-time from the preprocessed EEG signal. The PLL algorithm is a control system that estimates the EEG phase in real time at each sample point. Therefore, there is no time window used for phase estimation. Since high-amplitude, low-frequency slow waves predominate during NREM sleep, the PLL can lock in phase with the EEG signal even in presence of other frequency components. Therefore, the PLL was applied in real time over the recorded and notch-filtered data that was further pre-processed with the high-pass filter at 0.1 Hz, avoiding extra undesired non-linear phase shifts induced by pre-processing filters. When the estimated phase reached the target stimulation phase (45°), the EEG phase condition was satisfied. The PLL parameters were optimized to maximize the stimulation triggers in the SW up-phase (from 0° to 90° phase range) and increased the phase accuracy around the target phase.

To avoid waking the participants due to the auditory stimuli, the volume of the stimuli was adapted to the depth of sleep and stimulus related detected arousal. The initial volume was set to

52 dB (default value). If no arousal was detected after seven consecutive tones and a sufficient sleep depth was reached, the volume increased in small steps (1.5 dB) until it reached a predefined maximum volume (60 dB). When an arousal was detected during two consecutive stimuli, the volume decreased by 1.5 dB until a predefined minimum of 46 dB. In addition, when minimum delta power was not satisfied, the volume was set to the default value.

In parallel, the system featured three reinforcement conditions that disabled stimulations when: (1) the preprocessed EEG had a negative amplitude to prevent stimulations in the slow wave down-phase; (2) a minimum time of 0.5 s was not accomplished from the previous delivered stimuli to prevent multiple stimulations on one slow wave; and (3) the EEG amplitude was above 300 μV indicating the possible presence of artefacts.

Finally, two different stimulation approaches were implemented: (1) ON or continuous approach, where stimulations were continuously applied, and (2) ONOFF or windowed approach where stimulations were enabled for an interval of 6 s (ON window), and immediately thereafter disabled for another 6 s (OFF window). We selected this window length to be comparable to previous publications using an ON/OFF windowed approach with windows that had on average the duration of ~6 s[24,36].

**EMA on mobile phone**. Android Apps built with lambdanative[37] were used to collect phone data. Data collected by phone was directly transferred and stored in REDCap[38,39]. After each intervention night, participants completed in the morning an EMA regarding their perceived sleep quality, bedtime, device usability/blinding, sleepiness, and their momentary mood. At 2 pm they completed again an assessment of sleepiness and performed the PVT on the phone. At 8 pm they rated their sleepiness and their mood for the whole day. In this manuscript, we only report the results from two subjective sleep quality questions, the sleepiness at different time points during the day, the mood in the morning and at 8 pm, and the results of the PVT. Participants had to rate their overall perceived sleep quality on a 7-point Likert scale ranging from 0: *very bad sleep quality* to 6: *very good sleep quality*, and their overall perceived sleep depth on a 5-point Likert scale ranging from 0: *very superficial* to 4: *very deep*. Sleepiness was assessed using the KSS[40]. We used the "Mehrdimensionale Befindlichkeitsfragebogen" (MDBF, Steyer et al. 1997) to assess a summed mood score[41]. PVT (Dinges & Powell, 1985) is used to test attention, vigilance, and recovery from sleep[42]. For this study, we have developed a mobile PVT app using the touchscreen interface of the mobile phone with a mean maximal latency of 15.8 ms (sd 5.8 ms). During 10 min participants had to tap as fast as possible with their dominant index finger on a predefined area on the screen as soon as a counter appeared indicating the stimulus onset time in ms. Taps were excluded that had a reaction time (RT) equal or above 3000 ms or below 100 ms. We focused on three outcome measures that were indicated by Basner and Dinges (2011) to be most sensitive (e.g., most pronounced effect size) under sleep restriction conditions: mean speed (1/RT), fastest 10% reaction time, and slowest 10% of speed[43]. In this manuscript, we only included assessments of the days that had a good-quality/functional EEG assessment and stimulation the night before. Furthermore, assessments at 2 pm and 8 pm were only included if the start of the assessment was not deviating more than ±75 min from the scheduled time. Of note, on the last day after intervention, 2 pm and 8 pm assessments were consistently missing because home visits were mostly scheduled before 2 pm. A number of included subjects and assessments per EMA are summarized conditions in Supplementary Table S2. These EMA assessments of sleep quality, sleepiness, mood, and vigilance represent our secondary study outcomes.

**Hybrid sleep scoring**. Automated sleep scoring was performed for all nights using a deep learning classification that was followed by a manual expert review of classifications with low confidence scores. The Fpz-A2 derivation, sampled at 250 Hz, was pre-processed with a 0.5–38 Hz band pass filter. We designed the deep learning architecture as a fusion of two established sleep scoring neural networks adapted to work with each other. The first network used a sequence of convolutional layers followed by recurrent layers, as introduced by the DeepSleepNet[44], but altered such that the training is performed in a single stage and the network uses a many-to-many input-output relation. The second network replaced the convolutional layers with another set of recurrent layers as seen in the SeqSleepNet[45]. A sequence of 20 epochs was used as input for the models and by using a one stride in a many-to-many output scheme the network produced 20 predictions per epoch from each base model for a total of 40 individual predictions in the form of a likelihood distribution over all labels. The multiple predictions were combined into a single distribution through probabilistic multiplicative aggregation from which the final prediction was selected as the maximum likelihood label. To train the machine-learning algorithm, four expert raters scored manually a training set of 98 nights from the 24 subjects (7 subjects 9 nights each, and 17 subjects 2 nights each applying AASM sleep scoring guidelines[46] on 20-s epochs, however picking up all EEG information from the FPz-A2 derivation (84% inter-rater agreement). Both base networks were trained with the Adam optimization algorithm using a learning rate of 0.0001, a batch size of 32 for 50 learning epochs for the convolution-recurrent model and 10 learning epochs for the recurrent-recurrent model. Dropout was used for both networks with a probability of 50 and 75%. The final model selection was done through assessment of the accuracy on a left out validation set every 100 training steps. This model was then used to score the remaining nights. To further increase the scoring quality, a confidence score was assigned to each automatically scored epoch by comparing the likeliness of each predicted class. The N2 or N3 epochs that obtained low confidence scores (8.8%) were presented to an expert for rescoring while blinded to the predicted class. This manual re-scoring was done by an expert not involved in the study, and he was not informed about the conditions and intention of the study. This approach increased the classification accuracy of N2 and N3 in the training set from 75 to 78.8%.

All nights were excluded in the sleep architecture analysis that had either more than 20 min bad EEG signal quality, only partial recordings present or had electrodes not yet connected when recording started or disconnected before recording was stopped for more than 20 min. The number of included nights per condition are summarized in Supplementary Table S2.

**EEG analysis**. Before spectral analysis was performed, EEG was high-pass filtered at 0.5 Hz and low-pass filtered at 40 Hz using FIR filters provided by the EEGLAB toolbox[47,48] for MATLAB (The MathWorks, Inc., Natick, MA). Thereafter, the EEG was either analyzed with a consecutive analysis or ON-OFF analysis method. For both methods, spectral power of 6-s epochs was calculated using the MATLAB *pwelch* function with 4 s window length with 50% overlap. We analyzed SWA, that is EEG power in the slow wave frequency range, as our primary outcome variable to investigate the main effect of the auditory stimulation on brain oscillations. This is among the most consistently reported findings in previous in-lab studies using auditory slow wave sleep modulation (reviewed in Grimaldi et al.[27]). Furthermore, for the spectral density analysis in

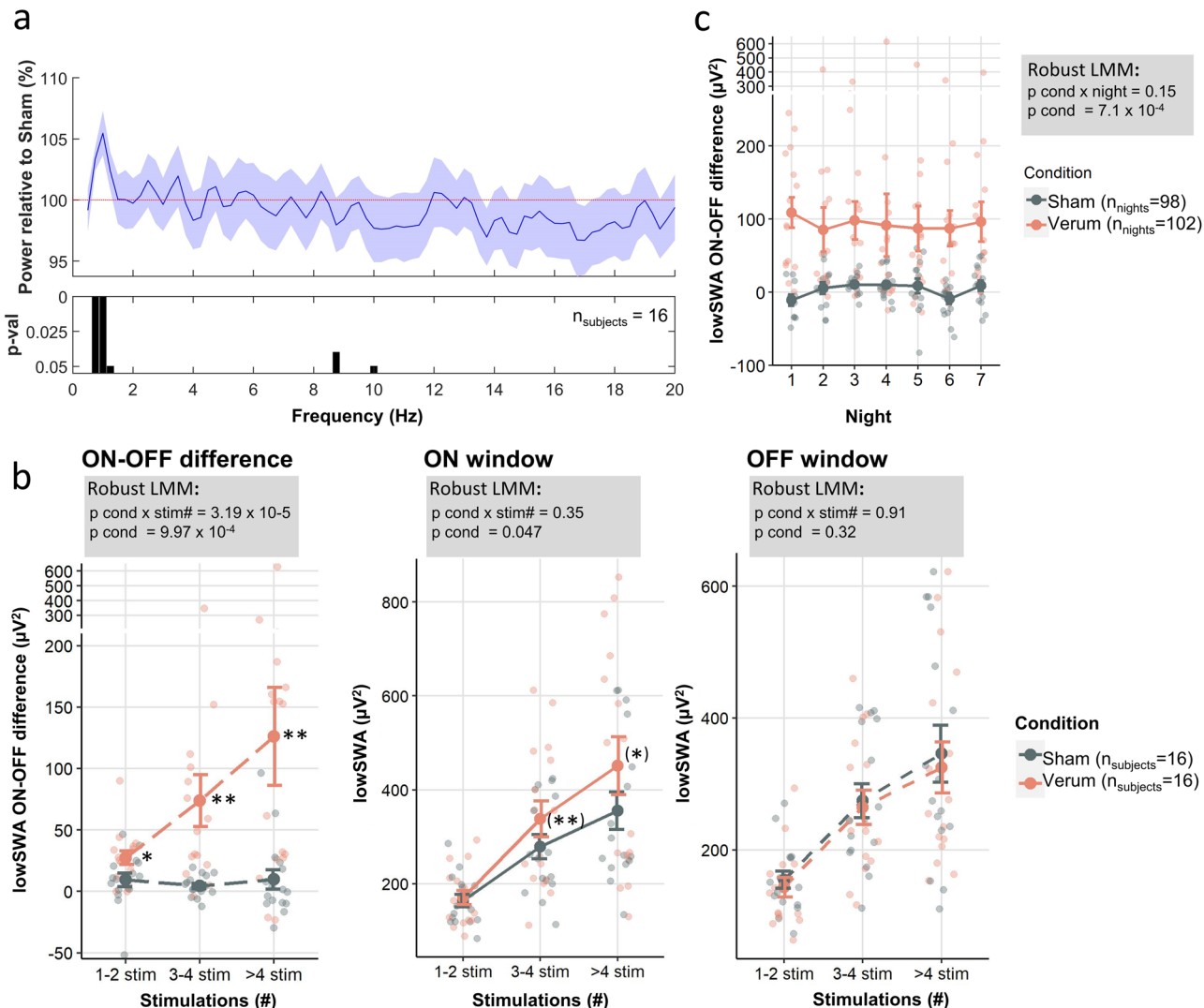

**Fig. 2 Slow wave effect of auditory stimulation in the windowed stimulation approach. a** Ratio (mean, shaded area SEM) between Verum and Sham in the windowed approach of the normalized spectral density (power divided by cumulative electroencephalography [EEG] power up to 30 Hz) during non-rapid eye movement (NREM) sleep (device and offline detected, consecutive analysis see methods) shows a significant cluster of frequency bins in the low-frequency range of 0.75–1.25 Hz (lowSWA). *P*-values (fixed factor condition, cond) were derived from a robust linear mixed model (see methods) performed for each frequency bin separately. **b** Error bars (mean ± SEM) with data points representing individual participants of ON-OFF window analysis of lowSWA for the difference between ON-OFF, ON windows, and OFF windows (see methods). Windows with stimulations (triggers or tones) were sorted and matched for their number of stimulations (stim#, 1–2 stimulations (1–2 stim), 3–4 stimulations (3–4 stim), and more than 4 stimulations (>4 stim)) and compared between Verum and Sham condition. Overall robust linear mixed model results (model includes average of nights) are summarized in the gray panels. The y-axis has been condensed to better visualize the error bars despite outliers. In case of significant main or interaction effects, a robust linear mixed-effects model was calculated for each stimulation bin separately (robust linear mixed-effect model [LMM], model includes all nights, *indicate *p* < 0.05 **indicate *p* < 0.01, stars in brackets illustrate exploratory analysis because of non-significant interaction effect). Effect of ON-OFF difference is driven by a significant increase of lowSWA in the ON window as indicated by a significant main effect of condition in a robustLMM, but not the OFF window. Following this significant overall model, separate robust LMM including all nights depict that specifically stimulation bins with more than 2 stimulations (3–4 stim, >4stim) show a significant condition effect in the ON window (*indicate *p* < 0.05). **c** In an additional analysis of all the ON-OFF window differences with more than 2 stims, we focused on whether there is a consistent difference between Verum and Sham for all seven nights. Results are shown as error bars (mean ± SEM) and data points represent individual nights of 16 participants. The y-axis has been condensed to better visualize the error bars despite outliers. In a robust LMM with factors nights, conditions, and control fixed and random factors (see methods) revealed no significant interaction night x condition (*p* > 0.1) and a main effect condition (*p* < 0.001), indicating a significant effect between Sham and Verum across all seven nights. Solid lines refer to ON window results, long dashed lines to ON-OFF difference and short dashed lines to OFF window results. Data underlying this figure is provided in Supplementary Data 1.

Figs. 2a and 3a, we further normalized the power spectra by its cumulative power up to 30 Hz as to be comparable with previous auditory stimulation studies[20,21] and to define our slow wave activity (SWA) frequency range of interest. For all other spectral-related analysis, absolute power values were used in the designated frequency band. For all spectral related analysis (ON-OFF analysis and consecutive analysis) only 6-s epochs that were recognized by both, the device algorithm and the hybrid scoring as N2 or N3 NREM sleep for the entire 6 s were included. Furthermore, in order to avoid outlier affecting the calculated spectral analysis, only epochs

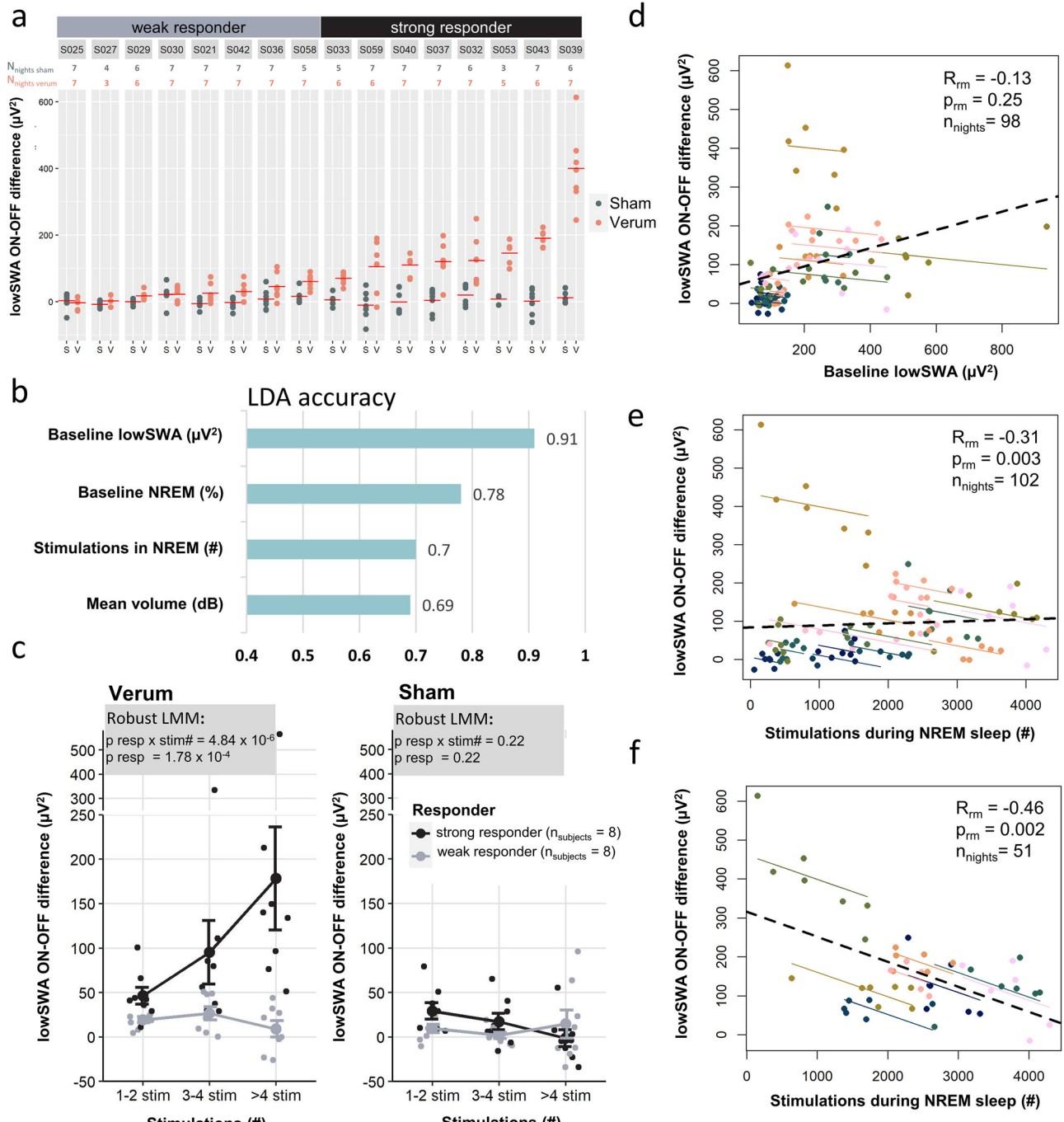

**Fig. 3 Predictive factors for across and with subject variability in ON-OFF difference in low-frequency slow wave activity (lowSWA) during the windowed approach. a** Individual (all nights of $n_{subjects} = 16$) ON-OFF difference (calculated using ON-OFF analysis, see methods) of lowSWA for all windows with >2 stimulations sorted by averaged lowSWA (red horizontal lines) during Verum. Strong responders refer to the upper 50% and weak responders to the lower 50% of participants. Clear ON-OFF differences during Verum condition across the majority of nights are seen for strong responders but not that clearly for weak responders. (S = Sham, V = Verum). **b** Linear discrimination analysis (LDA) using 10-fold cross-validation. All good quality nights during Verum ONOFF were included in the model and labeled as weak or strong responder night (dependent variable for classification). For lowSWA and %NREM N3 (non-rapid eye movement stage 3) during baseline, number of stimulations during NREM sleep and mean sound volume (prediction factors) a separate LDA model was run and accuracy in predicting weak and strong responder nights were plotted in blue. * indicate $p < 0.05$ comparing accuracy achieved by the prediction factors to chance level (No Information Rate). **c** Comparison of lowSWA ON-OFF difference between weak and strong responder (factor resp) for windows that only included sound volume at 52 dB (baseline lowest level) and for different stimulation bins (factor stim#) using a robust linear mixed-effect model (LMM) illustrated as error bars (mean ± SEM) with data points representing individual participants. The y-axis has been condensed to better visualize the error bars despite outliers. **d–f** Repeated measures (rm) correlations to investigate the prediction of nightly lowSWA ON-OFF difference variance. **d** Predictor baseline lowSWA for all good quality Verum ONOFF nights. **e** Predictor number of NREM sleep stimuli for all good quality Verum ONOFF nights. **f** Predictor number of NREM sleep stimuli for all good quality Verum ONOFF nights in strong responder subjects. Data underlying this figure is provided in Supplementary Data 2.

that were not identified as artifacts by the semiautomatic artifact rejection[49,50] and by an automatic outlier detection in the 6 s epochs (*isoutlier* function of MATLAB) were included. The ON-OFF analysis reported in Figs. 2b, c, d, and 3 was based on windows defined as ON or OFF by the MHSL-SB algorithm during the windowed approach nights. We considered the ON and OFF windows that entirely coincides with the hybrid scoring and MHSL-SB stable NREM sleep classification. Thereafter, we discarded windows with artefacts or high beta power detected by the MHSL-SB algorithm and windows without stimulation (or trigger in the case of Sham nights). To analyze whether there is a dose-dependent effect, the ON and OFF windows were binned and compared by the number of stimulations occurring during the ON and OFF windows (1–2, 3–4, and more than 4 stimulations). Thereafter, specifically windows with more than 2 stimulations were considered for analysis. The consecutive analysis method was applied in Figs. 2a, 4 and 5 for both the continuous and windowed stimulation approach to allow for comparability between approaches. Here, the whole recording was divided in consecutive 6-s epochs and for each epoch number of stimulations were defined. For Figs. 2a and 4a, c all epochs that entirely coincided with the hybrid scoring and MHSL-SB stable NREM sleep classification were included. For Figs. 4b, d and 5, epochs that fulfill the above criteria and had >2 stimulations were selected. Of note, spectral analysis of the robust NREM baseline (before stimulation started) was using consecutive 6-s epochs that entirely coincided with the hybrid scoring and MHSL-SB stable NREM sleep classification and had no artifacts (see above).

Auditory evoked responses of all stimuli (tones or triggers) were calculated. For the windowed approach these responses were analyzed for all ON and OFF windows and again differentiated the windows for their respective number of stimulations. We also calculated the ON-OFF difference by subtracting the average OFF ERP from the ON ERP for the respective stimulation numbers.

Arousal were automatically detected during hybrid scored NREM N1-N3 sleep and REM sleep using a published, open-source algorithm[51,52]. In short, the algorithm uses one raw EEG and EMG derivation and detects arousal-based EEG power in the alpha (8–12 Hz) and in the beta (>16 Hz) bands. Of note, all nights that had more than 20 min bad EEG signal quality or had only partial recordings present were excluded from the EEG statistical analysis. The number of included nights per condition is summarized in Supplementary Table S2.

**Statistics and reproducibility**. Statistical analyses were performed using the statistical software R[53] (R Foundation for Statistical Computing, Vienna, Austria). In order to establish the effect of condition, approach, and responder on sleep and daily functioning, we employed robust linear mixed-effect models (LMM) using the R-package *robustlmm*[54,55] for all interval-scaled variables. Robust statistical methods have the advantage that they provide accurate p-values even if some assumptions (e.g., normal distribution) are violated[56]. We used the Kenward–Roger approximation to estimate $F$, p-values, nominator, and denominator degrees of freedom[57]. For ordinal data obtained from the questionnaires (sleep quality, KSS), cumulative linked mixed models were employed using the R-package *ordinal*[58,59] and p-values obtained using the Likelihood Ratio Test. Depending on analysis in focus, different fixed and random factors were used. For Fig. 2a, b (post-hoc testing), and c fixed factors condition (2 levels: Verum, Sham), night (7 levels), period order (2 levels), and nested random factor of subject and condition were included. Because condition or the interaction of the factors condition and night were the focus of the analysis respective F and p-values were reported. In Fig. 2b, nights per condition were averaged and model included the fixed factors condition, number of

stimulations of the respective ON and OFF window (3 levels: 1–2 stimulations, 3–4 stimulations, >4 stimulations), period order, and nested random factor of subject and condition. For Fig. 3c, analysis was performed for Sham and Verum separately, averaged nights per condition were included with fixed factors responder (2 levels: weak, strong), number of stimulations, and the random factor subjects. For Fig. 4a fixed factors condition (2 levels: Verum, Sham), night (7), period order (2 levels), and nested random factor of subject and condition were included. Results reported in 4b, and in Table 1 and Supplementary Table S3 included the fixed factors condition, responder, night, period order, and nested random factor of subject and condition. In Fig. 4c, fixed factors approach (2 levels: ON, ONOFF), responder, night, period order, and nested random factor of subject and approach were included in the mixed model. Finally, Fig. 4c applies a model for responders only, for both Sham and Verum condition separately, including the factors approach, night, period order, and nested random factor of subject and approach were included in the linear mixed model. Subject S039 and all related nights were excluded from analysis that included the continuous approach because the participant had complete data loss during the Verum ON condition.

For the evoked response analysis in Supplementary Figs. S2 and S3 we implemented a LMM with fixed factors condition (2 levels: Verum, Sham), night (7 levels), period order (2 levels), and nested random factor of subject and condition for Supplementary Figs. S2a and S3a. In Supplementary Figs. S2b and S4b we focused on the interaction between responder and condition. Supplementary Fig. S3b and d included the fixed factors approach (2 levels: ON, ONOFF), night (7 levels), period order (2 levels), and nested random factor of subject and approach. Please note that non-robust statistics were used here due to too extensive computation time for this analysis. However, the p-values obtained were extremely small and independent of whether a robust or non-robust approach is used the outcomes would have been similar.

In order to classify whether lowSWA baseline and stimulation characteristics of each night during Verum are predictive for whether they belonged to a weak or a strong responder, we performed a linear discrimination analysis (LDA, Fig. 3b) using a 10-fold cross-validation with the R-package *caret*[60,61]. To establish whether the obtained accuracies were better than the No Information Rate (best guess without additional information) a one-tailed binomial test was performed.

To assess whether nightly variance within subjects of an outcome parameter of interest (e.g., lowSWA ON-OFF difference) was influenced by specific predictive factors (e.g., baseline lowSWA), we performed repeated-measures correlation using the R-package *rmcorr*[62,63]. This type of correlation takes multiple measures of participants into account and calculates the common intra-individual association between two measures. The package establishes the best linear fit for each participant's regression lines with the same slope but varying intercepts. This statistical approach was used in Figs. 3d–f and 5. Importantly, repeated measures correlation only considers the intra-individual associations and the provided r- and p-value refers to the individual fitted, solid lines in the presented figures and not on the dashed lines that represent the inter-individual association or group-level association. This analysis also highlights that it is important to distinguish inter- and intra-individual associations as they do not necessarily need to go in the same direction[62].

Overall, no corrections of multiple comparisons were performed, and we therefore specifically focused on effects that were consistently observable across approaches. P-values < 0.05 were considered significant and a p-value < 0.1 and ≥0.05 as trend-level (two-sided assessments, otherwise noted). Figures

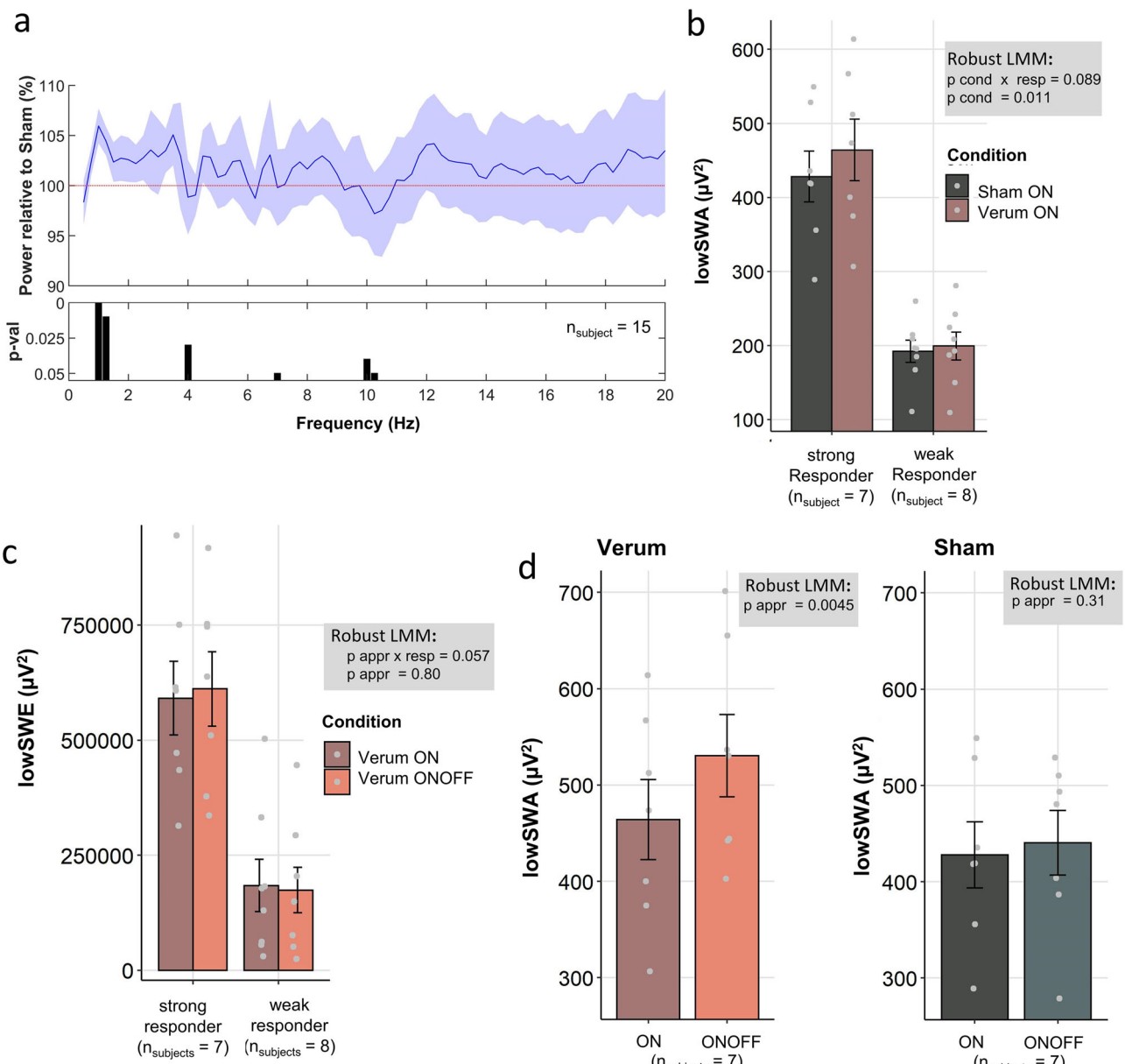

**Fig. 4 Slow wave spectral analysis in the continuous approach in comparison/relation to the windowed approach using the consecutive analysis method. a** Ratio (mean, shaded area SEM) between Verum and Sham of the normalized spectral density (power divided by cumulative EEG power up to 30 Hz) during non-rapid eye movement (NREM) sleep (device and offline detected) shows a significant cluster of frequency bins in the low-frequency range of 1–1.25 Hz. *P*-values (fixed factor condition) were derived from a robust LMM (see methods) performed for each frequency bin separately. **b** Slow wave activity in the low-frequency range of 0.75–1.25 Hz (lowSWA) in windows containing more than 2 stimulations for Sham and Verum, for weak and strong responders separately illustrated as error bars (mean ± SEM) with data points representing individual participants. Robust linear mixed model results comparing conditions (factor cond) and the interaction of the condition (Verum vs Sham) and responder (factor resp, weak vs strong) are depicted in the gray panel. **c** LowSWA summed overall device and offline detected NREM sleep periods (energy, lowSWE) for Verum ON (continuous approach) Verum ONOFF (windowed approach), for weak and strong responders separately illustrated as error bars (mean ± SEM) with data points representing individual participants. Overall robust LMM results (all night included in model) regarding approach (appr) and the interaction of approach and responder are summarized in the gray panel. **d** LowSWA in windows containing more than 2 stimulations comparing windowed and continuous approaches in responders ($n_{subjects} = 7$), for Verum and Sham separately illustrated as error bars (mean ± SEM) with data points representing individual participants. Robust LMM comparing approaches (continuous vs windowed) are summarized in the gray panel. Data underlying this figure is provided in Supplementary Data 3.

were done in MATLAB or in R using the R-package *ggplot2*[64,65]. For truncating and condensing the *y*-axis of figures with outlying values, the R-package *ggbreak*[66] was used.

Reproducibility: Our multiple nights of stimulation, and the two stimulation protocols that target the up-phase of sleep slow waves (just dose-dependent differences) provide robust findings

because (1) across multiple nights the effect on SWA was seen and (2) findings in windowed stimulation approach are reproduced in continuous approach. For the performed interim analysis that is reported here, our sample size was based on previously published lab studies that showed in a within subject design, a significant effect of auditory stimulation on slow waves.

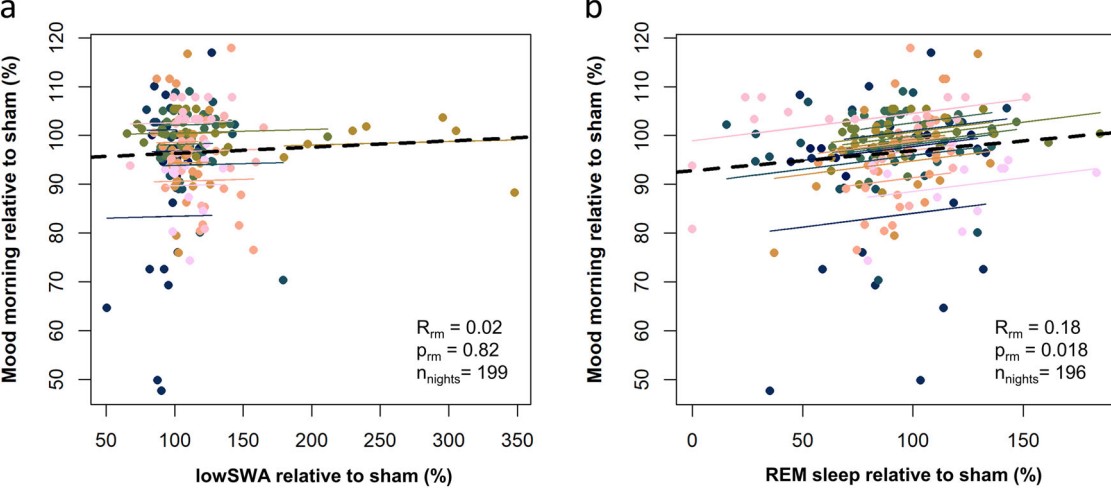

**Fig. 5 Repeated measures correlation of mood with low-frequency slow wave activity (lowSWA) and rapid eye movement (REM) sleep.** Repeated measures (rm) correlation of (**a**) slow wave activity in the low-frequency range of 0.75–1.25 Hz (lowSWA) (calculated using the consecutive analysis (see methods), windows with >2 stimulations) changes in Verum relative to Sham and (**b**) REM sleep changes relative to Sham with mood changes relative to Sham. Mood assessments were used from the morning assessments (lower values indicate a more negative mood). All nights during Verum were taken (pooled approaches in this correlation) relative to the averaged values of the respective Sham period (Sham ONOFF in windowed Verum nights, Sham ON in continuous Verum nights). Data underlying this figure is provided in Supplementary Data 4.

Several landmark papers had ~13 participants that reported a significant effect on sleep upon auditory stimulation in different age groups (e.g. Young: Ngo et al.[20]; Besedovsky et al.[17]; Older adults: Papalambros et al.[24]). We used 16 participants to match with this number and to allow for a counter-balanced study design.

**Reporting summary**. Further information on research design is available in the Nature Research Reporting Summary linked to this article.

## Results

**Experimental design**. Here we report the results of sixteen older adults (62–78 years, 7 female, enrollment overview in Supplementary Fig. S1) completing a six-week randomized, crossover experimental protocol in which they had two weeks of auditory slow wave stimulation (Verum) and two weeks of Sham interleaved with a two-week washout period (Fig. 1a). During Sham and Verum intervention periods, participants were self-applying and wearing the MHSL-SB, a mobile system for sleep-biosignal monitoring and real-time, phase-targeted auditory stimulation of slow waves at home. During Verum condition, MHSL-SB delivered short bursts of pink noise via on-ear earphones precisely timed during up-phases of NREM-sleep slow waves and their timing was saved. During Sham condition, toneless triggers were also saved. We divided the 2-week intervention periods into a windowed (ONOFF) and continuous (ON) application, each lasting for 7 days (Fig. 1b). The order of ONOFF and ON was counter-balanced. Here, we first consider the ONOFF windowed approach because it has previously been successfully applied to modulate slow waves as repeatedly demonstrated by changes in SWA[21,24,26,28,36]. Effects on SWA, therefore, represent our primary outcome. Furthermore, the windowed approach provides the opportunity to establish the within night effects by contrasting the ON to the OFF windows, to define weak and strong responders, and to reveal predictors for across and within participant variance. We then focus on the effects of the continuous application, which provides the unique opportunity to (1) establish robust effects on sleep SWA and sleep architecture that are seen in both, windowed and continuous approach, and (2)

elucidate whether more/continuous stimulation has additional benefits. Each day, participants completed multiple EMA with their mobile phone, including a sleep quality questionnaire, psychomotor vigilance test, the Karolinska sleepiness scale (KSS), and a mood assessment. In a final step, we establish whether auditory slow wave stimulation affects the behavioral/subjective assessments (secondary outcomes) for both, ON and ONOFF application.

**Windowed auditory modulation paradigm enhanced low-frequency slow wave activity over multiple nights in a dose-dependent manner**. There was a significant to trend-level increase in spectral power during detected NREM sleep in the low-frequency slow wave range between 0.75–1.25 Hz (lowSWA) comparing Verum to Sham in the windowed approach (Fig. 2a). In the next step, we specifically focused on the ON and OFF windows and lowSWA using the ON-OFF analysis method (see methods) to define whether there is a dose-dependent effect (Fig. 2b). In other words, ON and OFF windows were quantized by the number of stimulations occurring during the ON and OFF windows (stim#). Using a robust linear mixed model approach the difference between ON and OFF windows showed a significant interaction between a number of stimulations and conditions with a more pronounced difference for an increasing number of stimuli suggesting a dose-dependent effect ($F_{\text{cond} \times \text{stim\#} (2,54.2)} = 12.6$, $p = 3.19 \times 10^{-5}$). Moreover, a significant main effect of the condition was observed highlighting that Verum stimulation increased lowSWA in general ($F_{\text{cond} (1,12.7)} = 18.1$, $p = 9.97 \times 10^{-4}$) but more pronounced for windows with more auditory stimuli. Post-hoc analysis of each stimulation bin separately confirmed this finding, and all stimulation bins showed a significant difference to Sham using a robust mixed effects model approach including all nights in the model (1–2 stimulations: $F_{\text{cond}(1,12.4)} = 7.95$, $p = 0.015$; 3–4 stimulations: $F_{\text{cond} (1, 12.4)} = 32.1$, $p = 9.12 \times 10^{-5}$; >4 stimulations: $F_{\text{cond} (1,12.6)} = 11.5$, $p = 0.005$). However, from the ON-OFF difference, it remains unclear whether stimulation enhances lowSWA in the ON window, reduces it in the OFF window, or both. Therefore, we ran the same robust mixed effect models for ON and OFF windows separately. The ON-OFF difference was clearly driven by a significant increase of lowSWA in the Verum ON

**Table 1 Daytime assessments for the windowed and continuous approach.**

| | Windowed (ONOFF) approach | | | | | | Continous (ON) approach | | | | | |
| | Weak responder | | Strong responder | | statistics | | Weak responder | | Strong responder | | statistics | |
| | Sham | Verum | Sham | Verum | $P_{cond}$ | $P_{condxresp}$ | Sham | Verum | Sham | Verum | $P_{cond}$ | $P_{condxresp}$ |
| | Mean (SEM) | Mean (SEM) | Mean (SEM) | Mean (SEM) | | | Mean (SEM) | Mean (SEM) | Mean (SEM) | Mean (SEM) | | |
|---|---|---|---|---|---|---|---|---|---|---|---|---|
| **Sleep questionnaire** | | | | | | | | | | | | |
| Sleep quality[a] | 4.6 (0.34) | 4.56 (0.30) | 3.96 (0.25) | 3.87 (0.27) | 0.40 | 0.46 | 4.49 (0.33) | 4.34 (0.32) | 3.62 (0.33) | 3.91 (0.3) | 0.54 | 0.08* |
| Sleep depth[a] | 3.05 (0.25) | 2.96 (0.21) | 2.67 (0.16) | 2.52 (0.15) | 0.12 | 0.66 | 2.97 (0.22) | 2.76 (0.16) | 2.52 (0.11) | 2.43 (0.18) | 0.17 | 0.35 |
| **MDBF** | | | | | | | | | | | | |
| Mood morning[b] | 102.11 (3.10) | 98.47 (4.52) | 100.67 (5.44) | 96.79 (5.84) | 0.021* | 0.74 | 102.68 (2.76) | 100.27 (3.82) | 98.08 (5.94) | 94.83 (6.04) | 0.011* | 0.55 |
| Mood whole day[b] | 104.12 (2.45) | 96.39 (6.2) | 101.34 (4.94) | 98.99 (4.94) | 0.074* | 0.55 | 103.36 (2.7) | 103.31 (2.93) | 100.99 (5.09) | 95.36 (5.88) | 0.038* | 0.18 |
| **Sleepiness** | | | | | | | | | | | | |
| KSS moring[a] | 3.09 (0.24) | 3.07 (0.20) | 3.61 (0.22) | 3.76 (0.33) | 0.77 | 0.81 | 3.01 (0.18) | 3.1 (0.19) | 3.85 (0.3) | 3.8 (0.32) | 0.98 | 0.36 |
| KSS 2pm[a] | 3.04 (0.27) | 2.99 (0.28) | 3.29 (0.18) | 3.33 (0.39) | 0.94 | 0.71 | 3.1 (0.35) | 2.93 (0.28) | 3.21 (0.16) | 3.42 (0.27) | 0.96 | 0.48 |
| KSS evening[a] | 3.3 (0.30) | 3.14 (0.18) | 3.7 (0.20) | 3.87 (0.23) | 0.41 | 0.71 | 3.38 (0.38) | 2.85 (0.23) | 3.39 (0.21) | 4.12 (0.43) | 0.57 | 0.02* |
| **PVT** | | | | | | | | | | | | |
| Speed (1/RT)[b] | 3.14 (0.1) | 3.1 (0.1) | 3.05 (0.12) | 3.01 (0.09) | 0.17 | 0.86 | 3.15 (0.1) | 3.1 (0.09) | 3.07 (0.13) | 3.01 (0.08) | 0.50 | 0.97 |
| 10% slowest RT (ms)[b] | 378.55 (16.13) | 381.72 (15.04) | 401.86 (24.9) | 414.38 (18.56) | 0.22 | 0.85 | 377.96 (20.23) | 382.77 (15.93) | 406.02 (30.91) | 418.26 (16.84) | 0.80 | 0.47 |
| 10% fastest speed (1/RT)[b] | 3.53 (0.09) | 3.47 (0.10) | 3.51 (0.10) | 3.47 (0.08) | 0.029* | 0.73 | 3.51 (0.09) | 3.48 (0.08) | 3.52 (0.11) | 3.46 (0.08) | 0.77 | 0.88 |

Mood was assessed by the "Mehrdimensionaler Befindlichkeitsfragebogen" (MDBF). Sleepiness was assessed by the Karolinska Sleepiness Scale (KSS). PVT refers to a mobile version of the psychomotor vigilance task (10 min duration) to assess reaction time (RT). Cumulative link mixed model for ordinal data was used for all variables marked with a. Robust linear mixed models were performed to compare Sham to Verum nights for all interval scaled variables marked with b. P-value of factor Condition (cond, Sham vs Verum) and the interaction of condition and responder (resp, weak vs. strong) for each approach is reported in the statistics columns. Significant and trend-level p-values are marked with an asterisk (*).

window compared to Sham (main effect condition: $F_{cond\ (1,\ 12.7)} = 4.84$, $p = 0.047$, interaction condition and stimulation number: $F_{cond\ x\ stim\#\ (2,54.2)} = 1.08$, $p = 0.35$) but not by the OFF windows (main effect condition: $F_{cond\ (1,\ 12.7)} = 1.06$, $p = 0.32$, interaction condition and stimulation number: $F_{cond\ x\ stim\#\ (2,54.2)} = 0.09$, $p = 0.91$). Exploratory post-hoc analysis in the ON windows revealed that 1–2 stimulations did not significantly enhance low-SWA compared to Sham (main effect condition: $F_{cond(1,12.1)} = 1.24$, $p = 0.29$), but in the stimulation bins with more than two stimulations, ON had significantly more lowSWA during Verum than Sham periods (3–4 stimulations: $F_{cond\ (1,12.6)} = 9.17$, $p = 0.01$; >4 stimulations: $F_{cond\ (1,12.6)} = 7.53$, $p = 0.017$). Therefore, the lowSWA enhancement seems to be specifically robust when more than two stimulations are present, and we continued our analysis by focusing on these specific windows. These results were supported by auditory evoked responses (Supplementary Fig. S2a) indicating that the averaged time series across stimulations were significantly different between conditions and the extent of the differences were more pronounced with increasing stimuli number per window for the ON windows and the difference between ON and OFF response, but not for the OFF-window triggers only. Specifically, with more stimulations per window the response got more pronounced in the temporal domain illustrating that subsequent stimuli further enhanced slow wave amplitudes and therefore, indicating a cumulative effect and a dose-dependent enhancement of lowSWA. Finally, we investigated whether this lowSWA enhancement as indicated by the ON-OFF difference is consistently observable across 7 nights (Fig. 2c). There was no significant interaction of condition and night for the ON-OFF difference of all windows with more than two stimulations ($F_{cond\ x\ night\ (6,141.8)} = 1.62$, $p = 0.15$), the main factor condition was highly significant ($F_{cond(1,12.6)} = 19.76$, $p = 7.1 \times 10^{-4}$) indicating that the lowSWA enhancement was persistent across 7 nights.

**Within night enhancements of SWA in windowed approach showed pronounced inter- and intraindividual differences that were predicted by baseline lowSWA and stimulation properties.** Having identified that auditory stimulation can enhance low-SWA, specifically robust in windows with more than two stimulations, we identified individuals that show a consistent strong response and individuals that show weaker responses when comparing all Sham and Verum nights. Figure 3a illustrates the low-SWA ON-OFF difference for each subject and all nights during Verum and Sham conditions, sorted by the mean ON-OFF difference during Verum conditions. Strong responders are defined here as the upper 50% of participants and weak responders as the lower 50%. Strong responders represented in almost all nights a strong and clear difference from Sham. However, there are also weak responders that have smaller differences and more noticeable overlap with Sham. This was further confirmed by the evoked response analysis (Supplementary Fig. S2b) illustrating that the averaged difference between ON and OFF showed a more pronounced response for strong than weak responders during Verum but not during Sham. What predicts whether someone has a strong and consistent lowSWA response to stimulation or a weak one? In a first step, we compared the demographics between these two groups. Strong responders were significantly younger ($n = 8$, mean ± SEM: $65.5 \pm 0.8$ years) than weak responders ($n = 8$, $73.5 \pm 1.2$ years, independent samples Student's $t$-test, $t_{14} = 5.5$, $p = 7.8 \times 10^{-5}$, CI = $[-11.2, -4.9]$, Cohen's $d = -2.74$). There was no significant difference in the gender distribution of the two groups (strong responders: five females, weak responders: two females, Pearson Chi-square, $\chi 2(1) = 2.3$, $p = 0.13$) or difference in their hearing threshold (Pearson Chi-square, $\chi 2(2) = 2.8$, $p = 0.24$).

Older participants have less pronounced slow waves[13] and based on our algorithm stimulation logic, could likely receive less stimulations and at lower volumes, which could possibly explain group differences between strong and weak responses. Therefore, in a next step we performed a linear discrimination analysis to elucidate the probability of baseline lowSWA and stimulation properties in predicting whether the nights belonged to a strong or weak responder group (Fig. 3b). Before stimulation onset, the algorithm detected a 10-min window with robust NREM sleep and no stimulations occurring in which we calculated lowSWA. This baseline lowSWA predicted most accurately with 91% (Specificity = 1, Sensitivity = 0.82) nights of strong responders vs. weak responder (Fig. 3b). Participants with higher baseline lowSWA benefited stronger from stimulation than participants with less pronounced lowSWA (Supplementary Fig. S4). We further explored whether the amount of N3 NREM sleep (expressed as percentage) in the baseline could serve as a predictor in the LDA. It distinguished nights into weak and strong responders with an accuracy of 78% (Specificity = 0.88, Sensitivity = 0.69), which was highly significant but less predictive than with lowSWA. Overall, the main distinction of N2 and N3 NREM sleep is the number of slow waves with an amplitude of at least 75 μV and therefore we expect a clear overlap of the findings. We believe that lowSWA is a more sensitive marker because it represents a continuous and objective metric of sleep depth available throughout sleep, while distinguishing NREM sleep stage N2 and N3 are discreet classes with artificial thresholds. Therefore, baseline lowSWA could potentially be used to predict the stimulation success of individuals. However, one simple explanation of this finding could also be that based on our algorithm where stimulation and volume are dependent on delta thresholds (see methods), strong responders had more stimulations and louder acoustic stimuli. Both the number of stimulations and volume also significantly better discriminated strong responder vs. weak responder nights than chance level (Fig. 3b), yet their accuracy with 70% and 69%, respectively, were clearly lower than for baseline lowSWA. Furthermore, in order to elucidate whether strong responders enhanced lowSWA significantly better than weak responders independent from volume and controlled for the number of stimulations, we performed a mixed effects model analysis including only windows with acoustic stimuli at the default volume of 52 dB and for defined stimulation number bins (Fig. 3c). During Verum nights, strong responder had a more pronounced ON-OFF difference of lowSWA (main factor responder: $F_{resp\ (1,12.7)} = 27.35$, $p = 1.78 \times 10^{-4}$), and this difference increased with number of stimulations (responder × number of stimulation interaction: $F_{resp\ x\ stim\#\ (2,25.3)} = 20.63$, $p = 4.84 \times 10^{-6}$). These findings point towards more pronounced lowSWA modulation in strong responder vs. weak responders independent of volume and number of stimulations per window.

Overall, baseline lowSWA and stimulation properties significantly predicted weak and strong responding individuals. In a next step, we aimed at elucidating whether these factors also predicted nightly variances within subjects. Therefore, we performed repeated measures correlations, which estimates the common regression slope, indicating the association shared among individuals (Fig. 3d–f). Baseline lowSWA did not significantly predict lowSWA ON-OFF difference in all participants ($r_{rm\ (81)} = -0.13$, $p_{rm} = 0.25$, CI = [−0.34; 0.09], Fig. 3d) or in the strong responding individuals ($r_{rm\ (42)} = -0.13$, $p_{rm} = 0.42$, CI = [−0.41; 0.19]). Number of stimulations during NREM sleep significantly negatively predicted the within subject nightly variance in lowSWA ($r_{rm\ (85)} = -0.31$, $p_{rm} = 0.0037$, CI = [−0.49; −0.10], Fig. 3e), indicating that more stimulations resulted in less pronounced lowSWA ON-OFF differences in windows with more than two stimulations. The prediction was even more pronounced when only including strong responding

subjects ($r_{rm\ (42)} = -0.46$, $p_{rm} = 0.0018$, CI = [−0.67; −0.18], Fig. 3f). Repeated measures correlations for the mean volume was not significant in all subjects ($r_{rm\ (85)} = -0.08$, $p_{rm} = 0.44$, CI = [−0.29; 0.13]) and in strongly responding subjects ($r_{rm\ (42)} = -0.13$, $p_{rm} > 0.39$, CI = [−0.42; 0.18]).

**The continuous stimulation approach enhanced lowSWA but did not outperform the windowed approach.** In the previous paragraphs, we have focused on the windowed approach of stimulation, which allowed us for within night analysis due to ON-OFF specific analyses. In the following paragraph, we will center our attention on the continuous stimulation approach, in which all detected slow waves were targeted, and verified whether some of the previous findings could be reproduced with this approach. Here a consecutive analysis method was employed to define the 6-s epochs for spectral analysis for both approaches (see Methods). One participant (S039) was excluded from the continuous analysis because all recording nights during Verum ON were missing (also when comparing the two approaches the participant was entirely excluded in both approaches). NREM spectral density analysis revealed that there was a significant increase in spectral power in the lowSWA range between 1 and 1.25 Hz matching with the range in the windowed approach (Fig. 4a).

Because the continuous approach does not allow for an ON-OFF analysis, we specifically focused on lowSWA during detected NREM windows that also included more than two stimulations (consecutive analysis, see Methods). LowSWA was significantly enhanced in the continuous Verum condition compared to Sham ($F_{cond(1,\ 11.8)} = 9.01$, $p = 0.011$, Fig. 4b) and this condition effect tended to be affected by whether someone was a weak or strong responder ($F_{cond\ x\ resp(1,\ 11.7)} = 3.45$, $p = 0.089$, Fig. 4b), matching with the findings reported for the windowed approach. Furthermore, comparing the evoked responses including all stimulations, the averaged time series significantly differed between Verum and Sham (Supplementary Fig. S3a) and this difference was more pronounced in strong than weak responders (Supplementary Fig. S3b).

During the continuous approach, about double the amount of stimulations compared to the windowed approach were applied (Verum ON [mean ± SEM]: 3756.65 ± 549.5; Verum ONOFF: 1931.17 ± 282.39). More stimulations might lead to a more pronounced difference in lowSWA summed over the stimulation time applied. We, therefore, compared low slow wave energy (lowSWE) summed over all detected NREM periods between the continuous and windowed approach in the Verum conditions (Fig. 4c). There was no significant difference between these approaches (robust LMM, $F_{appr\ (1,11.5)} = 0.07$, $p = 0.80$), however, a trend interaction between approach and responder was observed (robust LMM, $F_{appr\ x\ resp\ (1,11.5)} = 4.46$, $p = 0.057$). Contrary to our hypothesis, lowSWE was on average increased in the windowed approach, specifically in strong responders, yet this was not significant. A possible explanation could be that the brain response to tones might be more pronounced in cases where breaks are introduced between the stimulation windows. We would therefore expect in strong responders that their averaged lowSWA in windows with more than two stimulations is more pronounced in the windowed than the continuous approach. Indeed, lowSWA was significantly higher in the windowed Verum condition than in the continuous one in strong responders (Fig. 4d, robust LMM, $F_{appr\ (1,4.4)} = 29.17$, $p = 0.0045$). No significant difference was observed between the Sham conditions (Fig. 4d, $F_{appr\ (1,4.2)} = 1.36$, $p = 0.31$). This was further underlined by the analysis of the evoked responses indicating that the time series for the windowed approach showed significantly higher amplitudes than the continuous approach

when comparing the Verum condition (Supplementary Fig. S3c) but not when comparing Sham (Supplementary Fig. S3d). Thus, the continuous approach does not outperform the windowed approach but rather a windowed approach might be more effective in slow wave enhancement, specifically in strong responders.

Finally, we wanted to investigate the effect of both stimulation approaches on sleep architecture and arousals (Supplementary Table S3) as primary control outcomes. Overall, sleep architecture was generally preserved across different conditions and only small changes were observed. Of note, consistent in both approaches, there was a tendency of less REM sleep during Verum than Sham condition (windowed approach: $F_{cond\ (1,12.1)} = 4.35$, $p = 0.059$, continuous approach: $F_{cond\ (1,11.6)} = 3.82$, $p = 0.075$). Considering the consistent interactions of responders and condition, sleep latency was showing a significant interaction, both in the windowed ($F_{cond\ x\ resp\ (1,12.2)} = 4.9$, $p = 0.047$) and continuous approach ($F_{cond\ x\ resp\ (1,\ 11.5)} = 8.55$, $p = 0.013$). Examining the averages across conditions and responders indicates that sleep latency was primarily reduced in the weak responders upon stimulation.

**Auditory stimulation reduced mood ratings in both, the windowed and continuous approach, which was predicted by REM sleep changes, but not lowSWA.** In a final step, we evaluated how stimulations affected our defined secondary outcomes that focused on daytime functioning, including subjective sleep quality ratings, mood, daytime sleepiness, and psychomotor vigilance performance (Table 1). Overall, only mood was consistently changed in both stimulation conditions and across different time points. Specifically, sum-score assessed in the morning was significantly decreased by a few points in Verum condition compared to Sham indicating a more negative mood during both, the windowed ($F_{cond\ (1,12.5)} = 6.95$, $p = 0.021$) and continuous approach ($F_{cond\ (1,11.6)} = 9.20$, $p = 0.011$). This finding was further confirmed during a second questionnaire round in the evening (assessing the mood during the whole day), for which the decrease was significant in the continuous approach ($F_{cond\ (1,11.8)} = 5.48$, $p = 0.038$) and trend-level in the windowed approach ($F_{cond\ (1,12.4)} = 3.82$, $p = 0.074$).

Mood was consistently affected by auditory stimulation, but is it related to the observed sleep changes? To answer this question, we correlated the relative mood changes from Verum to Sham to the sleep markers that were consistently changed in the two approaches. LowSWA during windows with more than two stimulations using the consecutive analysis method did not predict differences in mood in a repeated measures correlation analysis ($r_{rm\ (182)} = 0.02$, $p_{rm} = 0.82$, CI $= [-0.13;0.16]$, Fig. 5a). However, REM sleep changes were positively correlated to mood changes ($r_{rm\ (179)} = 0.18$, $p_{rm} = 0.018$, CI $= [0.03;0.31]$, Fig. 5b), indicating the more REM sleep decreases, the more mood decreases. Finally, REM sleep changes were not correlated to changes in lowSWA during windows with more than two stimulations ($r_{rm\ (179)} = -0.04$, $p_{rm} = 0.59$, CI $= [-0.19;0.11]$), potentially indicating that the effects of stimulation on slow waves might be independent of the REM sleep and mood changes.

Collectively, a reduction in mood changes was consistently found across approaches and in two separate daytime assessments. These changes where not related to lowSWA changes but partially explained by a small, but consistent, reduction in REM sleep across approaches.

## Discussion

In this paper, we provide the first insights into the effectiveness of auditory deep sleep stimulation for older adults in the field. We show that by targeting the up-phase of slow waves we can significantly enhance deep sleep activity on the group level over multiple days. To the best of our knowledge, we are the first to show in a randomized cross-over clinical trial that auditory stimulation outside the well-controlled lab setting is efficacious in enhancing lowSWA in older adults. Even though we had a small sample size due to COVID-19 related reasons, this finding is specifically robust because we reproduced it in two different stimulation approaches, windowed and continuous, in the same individuals over multiple consecutive nights. Within-night analysis in the windowed approach further provided some indication of a dose-dependent effect illustrating that windows with more stimulations (e.g., 1–2 stimuli vs >2 stimuli) resulted in more pronounced enhancement of lowSWA on the group level. The evoked response analysis further underlined that more stimuli produced a more cumulative effect in time such that within a window more amplitude enhancements were found because of more subsequent stimuli. Windows with more than two stimulations might either represent successive (e.g., consecutive trains) or distributed targeting of slow waves within these 6-s windows that could potentially differentially affect the effectiveness in enhancing SWA. In our sample, within and across nights, we presumably have a mixture between these scenarios and a clear distinction is not possible. Future studies are needed that systematically apply stimulations of consecutive trains or distributed slow wave targeting and test the effectiveness thereof. Future studies are needed that strategically apply these scenarios and test the effectiveness thereof. Several studies in young participants have confirmed the efficiency of enhancing slow waves in controlled single session laboratory set-ups, and initial evidence has been published indicating that commercial in-home devices can enhance slow waves upon auditory stimulation[32,33,67]. Along these lines, previous in-lab studies have confirmed the feasibility of inducing slow oscillations or enhancing slow waves in middle-aged and older participants[24,29,35], however, their susceptibility to auditory stimulation was reported and discussed to be much reduced compared to younger adults[32,35,68]. Our analyses provide first insights into the deciding factors of their susceptibility to auditory stimulation.

To date, effects of auditory stimulation in individuals over 40 years of age were restricted to group-level analysis leaving it unclear whether reduced or non-existing benefits on slow waves were observable across all participants or driven by single individuals. In this work we took a closer look at inter-individual differences in susceptibility in a restricted age group (62–78 years) to better understand who benefits from auditory stimulation and what could be factors predicting its efficacy. Our unique study design was ideally suited to tackle this question because each stimulation condition and approach had multiple nights to allow for a robust separation of participants with weak and strong responses across multiple measurements. Specifically, the windowed approach enabled the analysis of within-night responses. Based on the nightly response in the windowed approach we divided our subjects into weak and strong responding individuals. These two groups substantially differed in the efficacy to enhance lowSWA upon auditory stimulation in the windowed approach, and we confirmed a tendency of more pronounced enhancement in strong vs weak responders also in the continuous approach, supporting the robustness of the finding. Interestingly, whether a person is a weak or strong responder is likely a trait characteristic, as illustrated in Fig. 3a. Thus, people that strongly respond, do so in the majority of the nights. Baseline lowSWA during NREM sleep was the best predictor, classifying with 91% accuracy whether the night belonged to a weak or strong responder. Arguably, individuals with more slow waves will also have more and louder stimulations as per the design of the stimulation algorithm,

possibly explaining more efficacious responses. However, (1) with accuracies around 70%, mean volume and number of stimulations were less well predicting weak and strong responders than baseline lowSWA, and (2) strong responders still showed a significantly more pronounced response when controlling for volume and number of stimulations (Fig. 3c). The weak responders were also significantly older than the strong responders, and with increasing age, there is a decline in the extent and amount of slow waves[13]. Yet, the difference is only a few years and even such a small age difference translates into a pronounced difference in how efficacious the stimulation is. Importantly, differences in the hearing thresholds unlikely explain the difference in efficacy because there was no significant difference between the groups. There might be a critical level of slow waves present for the stimulation to be efficacious. Along these lines, an alternative interpretation of the dose-dependent effect observed in Figs. 2 and 3c could be that not necessarily more stimuli are more efficacious, but rather more pronounced SWA levels enable more efficient responses. However, with our study design, we cannot disentangle whether the extent of slow waves is functionally relevant for them to be modulated or whether SWA just mirrors age-related changes in the susceptibility of the brain to react to sensory stimuli, e.g., a reduced integrity of the central nervous system[69]. Overall, this finding has important clinical implications because patients with neurodegenerative disorders, such as Alzheimer's and Parkinson's disease, would likely benefit from slow wave enhancement, but their sleep is characteristically hallmarked by diminished slow waves compared to age-matched healthy controls[18,70]. Therefore, susceptibility could be further reduced. It is therefore of outmost importance to establish the effectiveness of stimulation in these patients and consider prioritizing stimulation at pre-clinical stages, i.e. in mild cognitive impairment patients[36].

Besides inter-individual differences in the effectiveness of stimulation also intra-individual differences were observed, most pronouncedly in responders. Such a variability highlights the importance of clinical trials that assess effects over multiple nights and that single-session studies typically done in the lab might not reveal the complete picture. Are baseline lowSWA or stimulation characteristics also predictive for this nightly variance? No, only the number of stimulations predicted the effectiveness to modulated lowSWA in the windowed approach, specifically highly significantly in the strong responders. Against our expectations, the correlation was negative, indicating that more stimulations resulted in less pronounced lowSWA ON-OFF differences within individuals. This observation might relate to the literature showing that the brain might react more pronouncedly to repetitive sounds if they are rarer (e.g., longer breaks in between)[71,72]. Moreover, Ngo and colleagues[73] have previously shown that driving sounds are a self-limiting process and driving stimulations (e.g., as many as possible for consecutive slow oscillations) did not outplay a 2-click paradigm. Thus, there is reduced effectiveness in prolonged sequences of acoustic stimuli (trains) that might be explainable by a refractory period that builds up with consecutive auditory stimuli[73]. In this case, we would also expect that our windowed approach would elicit more pronounced responses than our continuous approach, for which about double the amount of stimuli was applied. Indeed, in strong responders, lowSWA was more pronounced during Verum in the windowed approach than in the continuous approach. Our results therefore point to more effective enhancement of SWA in the windowed approach if we only consider short-term timeframes, such as the 6-s windows. Of note, averaged epochs do not provide any information regarding the overall enhancement of lowSWA across the stimulation period. Investigating the cumulated lowSWA (lowSWE) over the whole detected NREM period, there was

no significant difference between the continuous and windowed approach emphasizing that at this stage we cannot term one of the approaches as superior. Interestingly, this finding seemingly contradicts the idea that there is a dose-dependent improvement, because lowSWE is not more enhanced in the continuous approach despite having around double the number of stimulations compared to the windowed approach. Yet, we suppose that a dose-dependent effect might specifically be observable in a short timeframe of a few seconds but in the perspective of a whole night more stimulations might reduce responsiveness. Thus, it is likely a trade-off between finding the ideal amount of stimulations and obtaining sufficient breaks between stimulation trains to maximize effectiveness in enhancing lowSWA and lowSWE across nights. This should be systematically investigated in future studies.

The decision in selecting an appropriate stimulation approach and target group also requires a careful consideration of secondary effects. We, therefore, quantified the impact of multiple days of auditory stimulation on sleep architecture, subjective sleep quality, and daytime functioning. To avoid overinterpretation of single significant results we focused here on results that were consistently changed across conditions and therefore illustrated a robust effect. Overall, only small differences were detected in sleep architecture, among which only REM sleep tended to be reduced in both stimulation approaches. This is in accordance with previous lab-based studies reporting no or only minor but inconsistent effects on sleep architecture[20,24,28,29]. Also sleep latency was reduced by a few minutes upon auditory stimulation in the windowed approach and a significant interaction was found between responders and condition for both approaches, specifically with weak responders showing a decrease. Were there any consequences on subjective parameters and daytime functioning? Neither subjective sleep quality, nor daytime sleepiness or vigilance was consistently affected by condition in both approaches. This is in line with previous in-lab studies showing that under normal sleep conditions (e.g., no sleep restriction), auditory stimulation did not affect them[20,24,35]. These assessed parameters might not be sensitive to slow wave changes in a healthy and sufficiently sleeping population. In contrast, effects might be expected if populations with insufficient sleep are targeted[67]. Among all daytime functioning assessments, only mood was consistently changed across conditions in both approaches and two assessment time points, which was partially related to the reduction in REM sleep. Associations between REM sleep and mood have previously been reported[74,75], we further provide initial evidence that REM sleep changes due to auditory stimulation might negatively affect mood. However, whether REM sleep causally influences mood needs further investigation. Importantly, neither REM changes nor mood changes were significantly correlated to lowSWA, the primary effect of auditory stimulation during sleep. Thus, secondary effects can occur that might be independent of the enhancement of slow waves and such possibilities need to be considered in the planning of future studies. A limitation of our study regarding daytime functioning is that we were possibly underpowered regarding assessments of daily functions. Furthermore, several assessments have not been consistently performed by participants or had to be excluded. Future studies that extensively assess secondary outcome parameters on a large scale, across different populations and sleep conditions (e.g. sleep restriction) are needed to better understand un-intentional side-effects of the stimulation. This is of great relevance considering (1) the raise of commercial solutions that promote sleep stimulation at home[32,33], and (2) clinical implications for in-home sleep optimization and treatment.

In conclusion, in this first in-home clinical trial we robustly show that SWA in the low-frequency range can be enhanced in

healthy older adults over multiple nights in a real-life setting. We illustrate that there are strong and weak responders, which represents a trait characteristic and is mainly predicted by baseline SWA. We further highlight that more stimulations are not necessarily more beneficial and that future rational design of auditory deep sleep stimulation should consider the optimal implementation of breaks to maximize overall deep sleep enhancement. While this study represents an example of a successful translation of in-lab findings to uncontrolled environments in a population with reduced slow waves, it also raises concerns about possible un-intended secondary effects that are independent of the modulation of deep sleep, such as a slight reduction in REM sleep along with a reduced subjective mood rating the next day. These findings highlight the importance of conducting future large-scale studies to establish secondary effects across time and diverse populations. Our study provides means to optimize the rational design of auditory deep sleep stimulation and represents a crucial step towards successful and safe in-home implementation for prevention and therapy in the general population.

## Data availability

Source data for the main figures in the manuscript can be accessed as Supplementary Data 1–4. Raw data that support the findings of this study cannot be made publicly available to protect participants' rights according to Swiss human research law. The de-identified individual participant data that underly the results of this paper can be accessed by investigators who (1) submit a methodological sound proposal describing the intended analysis and as reviewed by the authors of this publication, (2) provide proof of relevant ethical approval for the intended analysis, and (3) fulfill data protection measures according to Swiss legal requirements. The analyses of the shared data is restricted to achieve the aims of the intended analysis. Along with the data, no other documents will be made available. Proposals with reasonable requests may be submitted to walter.karlen@ieee.org up to 24 months following article publication.

## Code availability

The sleep classification model is based on previously published architectures which are publicly available at SeqSleepNet (https://github.com/pquochuy/SeqSleepNet) and DeepSleepNet (https://github.com/akaraspt/deepsleepnet).

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

## Acknowledgements

This work was conducted as part of the SleepLoop Flagship of Hochschulmedizin Zürich funded in part by the Schweizerische Hirn Stiftung, Muriel Nikles and the ETH Foundation, and the Swiss National Science Foundation (P3P3PA_171525 and PZ00P3_179795 to C.L.). The authors thank all the participants for taking part in the study and R. Büchi, N. Demarmels, G. Hoppeler, P. Kandasamey, J. Kurz, S. Schwander, and E. Silberschmidt for performing the recruitment and data collection. We also thank Rafael Wespi for his support in sleep scoring. SleepLoop consortium members provided uncountable discussions and feedback. We are grateful to the Senior University and URPP Dynamics of Healthy Aging at UZH for their support in participant recruitment for this study.

## Author contributions

The study was designed by C.L., R.H., and W.K. C.L., M.L.F. and W.K. developed the mobile intervention device. C.L., M.L.F., S.H., E.W., R.H. and W.K. participated in the acquisition, and/or interpretation of data. C.L., M.L.F. and L.B. analyzed the data. C.L., M.L.F. and W.K. wrote the manuscript and all co-authors reviewed and edited the manuscript, before approving its submission.

## Competing interests

C.L. is a member of the Scientific Advisory Board of Emma Sleep GmbH, which is not related to this work. R.H. and W.K. are founders and shareholders of Tosoo AG that has licensed the technology used in this work. All others have no competing interests to declare.
