## [Peer Review File · Communications Medicine]

Reviewers' comments:

Reviewer #1 (Remarks to the Author):

The study compares in healthy elderly the effect of two 7-day auditory phase-locked stimulation approaches to a 7-day baseline time and finds a robust enhancement in lowSWA power (0.75-1.25 Hz). The study is conducted in the home environment, showing feasibility and maintenance of the stimulation effect across the 7-day periods. A similar study, which the authors cite (Debellemannier et al 2018) had similar findings when investigating young and middle aged subjects. In the present MS the effect of the windowed approach (vs. a continuous approach) is produced most strongly by the stimulation of more than two waves within a 6 sec window. It is, however, unclear whether these are > 2 successive oscillations, or >2 oscillations distributed sometime within the 6 second window? This differentiation needs to be made, as the interpretation differs. In the latter case, it could simply mean that more slow waves (deeper NREM sleep) dominated the 6 sec window. Although emphasis is on EEG power, the MS would benefit by depicting evoked potentials, especially for the dose-dependent results.

The authors also investigate inter- and intra-individual relationships on the lowSWA enhancing effect. A predictive quality of lowSWA for night enhancements of SWA is shown to be dependent upon age. However, the argument that lowSWA enhancement in strong vs. week responders depends on baseline lowSWA requires refinement (as detailed below). Together, the study contributes to the transition "from bench to bedside".

l. 132, It suffices to say: "In other 131 words, ON and OFF windows were quantized by the number of stimulations occurring during the ON and OFF windows." Introducing "bins" without defining them impairs comprehension. Why not use "Stim No." or similar as in the statistical test?

l. 135, throughout: Is it a journal requirement to give the repeat the factors after "p"? They are already given with the F-value.

l. 150, add that these post-hoc analyses were exploratory since the superordinate ANOVA did not indicate a significant interaction.

Fig. 2B, middle, Since the interaction cond x stim is not significant, the asterisks should at most be gray to indicate this. It is misleading to treat significant post hoc tests without an ANOVA effect the same as those revealing a significant effect in the variance analyses. Also, I suggest using two asterisks for 3-4 stim, and only one for >4 stim in agreement with the p-values ($p < 0.01$ and $p < 0.05$).

l.187, "Before stimulation onset, the algorithm detected a 10-min window with robust NREM sleep and no stimulations occurring in which we calculated lowSWA." Since the criteria for the first stimulation were a 10-min window with robust NREM sleep there is no need to write: "no stimulations occurring" in the sentence.

l. 190-192. Although the Figure S2 looks convincing, information on subjects should be accompanied by their sleep stage, i.e., whether in N2 or N3. In addition, age should be given in the same graph. Could it be that older subjects had stable NREM with less N3? (cp. Also l.339-l.340)

l. 213-223, please make the difference between rm and pm in text, and Rm and Pm in Fig. 3D clear. What exactly does each express? More information in the legend would also be helpful.

l. 248 & fig. 4c: the inset of Fig. 4C should read ">0.1" (as in the text), not ">0.01".

l. 250-251 "... lowSWA was on average increased in the windowed approach, specifically in strong responders." This statement is an overemphasis, as only the interaction, but not post hoc tests reached significance.

l.255-256, Fig. 4D, it would here also be important to see during which NREM sleep stage stimulation occurred, as sleep stage itself may affect responsiveness (e.g., Genzel, doi: 10.1016/j.tins.2013.10.002).

I. 278-279 & Fig. 5, I assume decreased mood score corresponds to a more negative mood, but this should be explicitly stated in the results or figure legend.

I.291-293, "Finally, REM sleep changes were not predicted by changes in lowSWA during windows with more than 2 stimulations ($r_{rm} (179) = -0.06$, $p_{rm} > 0.1$, $CI = [-0.21; 0.09]$), further indicating that the effects of stimulation on slow waves are independent from the REM sleep and mood changes." Based on one analysis, it seems a bit far-fetched to conclude that the change in lowSWA had no impact on REM sleep. Limit the result to no correlation between these two measures (cp. also 412-413).

I.314-315, for reasons given above it is misleading to conclude that "windows with more stimulations resulted in more pronounced enhancement of lowSWA on the group-level." As it implies an increasing number of stimuli are beneficial. The main finding lies in the comparison of 1-2 vs. >2.

I. 500-501, were signals recorded as DC, or was there a hardware high pass filter? Which frequency was any low pass filter at recording?

I.507-508, please specify the direction of spectral power used to define NREM sleep. I imagine that both high OR low delta are high, and beta low? Give the frequency ranges here for all three bands.

I. 515, Are the 3 measures of sleep classification of performance metric given in percent? Please define these metrics.

I.522, as different definitions of phase are used in the field, a figure of a slow wave with phases given is necessary. Supply the information whether 0° corresponds to the maximal down-state value or to the turning point from down-to-up state?

I. 524-525, how long is the time window used to estimate EEG phase? Does preprocessing (before application of the algorithm) consist of any procedure other than 0.1 Hz high-pass filtering? - This is especially relevant if the manuscript of Ferster et al will not be published as open access.

I. 508, Were the predefined threshold values individualized? Were fixed quotients, e.g., of low delta to beta used?

I. 543, Fig.1, What is the rationale behind using 6 sec for the windowed approach?

Fig. 1, "detected"? Isn't "targeted" meant?

I.574, add that the values in Table S3 are per condition (as given in I. 608).

Fig. 1, I.623, add a graph showing which EEG analyses periods are used for each of the 2 approaches, like Fig. 1B for recording. It is confusing, e.g., in I. 623 (analyses) to refer to: ON-OFF analyses AND consecutive sec analyses, since recording conditions are the same. Are 6 sec FFT epochs of the continuous approach meant with the latter?

Fig.2B, C Similar to the above, additional color/line style coding would be nice to assist in distinguishing comparisons of ON-OFF periods in the windowed stimulation condition (Fig. 2B, left, Fig. 2C) from the comparison ON- (OFF-) window period of stimulation and sham of the windowed approach Fig. 2B (middle, right)

I.681, "was influenced" by ... is missing.

Inconsistent use of terms: I.385: "lowSWA power (lowSWE)", yet power is depicted in figures aside from Fig. 4C.

Three references are duplicated: 24 & 35, 25 & 37, 26 & 57.

Reviewer #2 (Remarks to the Author):

This is a well written, very practical report evaluating the effects of multi-night delivery of phase locked acoustic stimulation during deep sleep. The particular strengths are the multi-night data and the very careful analyses of factors determining inter-individual differences in responses to such

stimulation : number of stimuli per window, total number of stimulations, baseline lowSWA and and the effects of mood.

I liked the analyses comparing whether stimulation effects were due to ON or OFF windows and the addition of the analysis of mood change.

There isn't much to comment on this polished but disappointing set of results other than to point out the average regression/ multi night correlation dashed line going in the opposite gradient direction as the rest of the data.

The authors are to be commended on a fine contribution.

Reviewer #3 (Remarks to the Author):

In this manuscript, the authors have examined the auditory stimulation for elderly at home. The novelty of this work lies in the stimulation being applied at home using their portable monitoring/feedback-controlled modulation device. The authors have examined the effect of stimulation in sham and verum conditions. They found an enhancement of slow-wave activity after stimulation. However, there was an adverse effect on the duration of REM sleep and subjective mood. Overall, the manuscript is scientifically sounds and is very well-written.

It is unfortunate that the number participants had to be limited to 16 due Covid. Nonetheless, the effects appear to be robust even with a small sample size.

Typo on line 359 - yAlzheimer's -> Alzheimer's

Reviewers' comments: Point-by-point reply

Reviewer #1 (Remarks to the Author):

The study compares in healthy elderly the effect of two 7-day auditory phase-locked stimulation approaches to a 7-day baseline time and finds a robust enhancement in lowSWA power (0.75-1.25 Hz). The study is conducted in the home environment, showing feasibility and maintenance of the stimulation effect across the 7-day periods. A similar study, which the authors cite (Debellemannier et al 2018) had similar findings when investigating young and middle aged subjects. In the present MS the effect of the windowed approach (vs. a continuous approach) is produced most strongly by the stimulation of more than two waves within a 6 sec window. ...

The authors also investigate inter- and intra-individual relationships on the lowSWA enhancing effect. A predictive quality of lowSWA for night enhancements of SWA is shown to be dependent upon age. However, the argument that lowSWA enhancement in strong vs. week responders depends on baseline lowSWA requires refinement (as detailed below). Together, the study contributes to the transition “from bench to bedside”.

We thank the reviewer for their valuable and thorough feedback, and we highly appreciated the constructive inputs. Overall, we believe that with the points raised by the reviewer, the manuscript has clearly improved in clarity for the reader. We performed additional analysis and recognized in the course of re-analysing that there was a non-relevant miscalculation of the baseline window time. This very mildly affected some of the stimulation results. After correction we did not see any major changes that affected our presented results and conclusions. The changes in the text resulting from the re-calculation are further highlighted in green, all affected figures have been adapted.

Please find our statements to the individual points below:

It is, however, unclear whether these are > 2 successive oscillations, or >2 oscillations distributed sometime within the 6 second window? This differentiation needs to be made, as the interpretation differs. In the latter case, it could simply mean that more slow waves (deeper NREM sleep) dominated the 6 sec window.

We thank the reviewer for bringing up this discussion. In our stimulation approach we targeted all waves that were tracked by the phase tracking algorithm to have reached the degree criterion (after sleep and slow wave threshold were reached). In theory, both cases described by the reviewer are possible. Consecutive trains are presumably more likely in N3 stage than in N2 stage (e.g., more “isolated” slow waves) and we expect that in our sample a mixture of both scenarios has occurred. How and whether these two scenarios differentially affect the stimulation effectiveness is unclear. Considering that each participant has a different distribution of N2 and N3 and a mixture of various frequencies that define slow waves (e.g., different temporal intervals between tones due to frequency difference of slow waves), we believe that an objective and precise distinction of these two scenarios is likely not feasible in our data set. Future studies might be needed that employ stimulation paradigms pre-programmed specifically for successive or distributed stimuli application.

Taking this important consideration into account we added the following small paragraph to the discussion:

“Windows with more than two stimulations might either represent successive (e.g., consecutive trains) or distributed targeting of slow waves within these 6s windows that could potentially differentially affect the effectiveness in enhancing SWA. In our sample, within and across nights, we presumably have a mixture between these scenarios and a clear distinction is not possible. Future studies are needed that

strategically apply stimulations of consecutive trains or distributed slow wave targeting and test the effectiveness thereof.”

Although emphasis is on EEG power, the MS would benefit by depicting evoked responses, especially for the dose-dependent results.

We thank the reviewer for this great idea. We performed extensive analysis on the evoked responses (ERP) that helped to further confirm and inform our previous results of this paper. In a first step, we calculated the ERP of all stimuli (tones or triggers) for all ON and OFF windows and again differentiated the windows for their respective number of stimulations. We also calculated the ON-OFF difference by subtracting the average OFF ERP from the ON ERP for the respective stimulation numbers. We implemented a linear mixed effects model with fixed factors condition (2 levels: verum, sham), night (7 levels), period order (2 levels), and nested random factor of subject and condition were included for Figures 1A and 2A below. In Figures 1B and 2B we focused on the interaction between responder and condition. Figure 2C and D included the fixed factors approach (2 levels: ON, ONOFF), night (7 levels), period order (2 levels), and nested random factor of subject and approach. Please note that only non-robust statistics were used here due to too extensive computation time for this analysis (1 comparison ~1min, considering that each evoked response has 1500 datapoints it would not be feasible). Nevertheless, the p-values obtained were extremely small that independent of whether a robust or non-robust approach is used the outcomes would have been similar. The results are summarized in the Figure 1A below. We clearly see that independent of number of stimulations there was a significant amplitude increase of the positive and negative amplitudes of the stimulated wave compared to sham, but only for the ON and not in the unstimulated OFF window. Furthermore, more tones presented in a window resulted in more pronounced changes in amplitudes not only after the first second of tone onset but also before time 0 (stimuli onset). Those changes before time 0 can be explained because we included every tone or trigger in this analysis and therefore, slow wave enhancements before tone/trigger onset (time 0) are possible if another tone preceded the analysed one. Thereby, highlighting that more tones may lead to slow wave amplitude changes beyond the target wave. These multiple enhanced waves with increasing stimulation number are likely explaining the dose-dependent differences we see in lowSWA in Figure 2B. Thus, each tone initiates an enhanced amplitude of the stimulated wave and more stimulated waves sum up to more pronounced lowSWA changes. Thereafter, we also wanted to see whether strong and weak responders differed in their ERP. To do so, we looked at the ON-OFF difference of all stimulated waves separately for weak and strong responders. We can depict in Figure 1B that the evoked response is more pronounced for strong than weak responders as indicated by a significant condition and responder interaction in the ERP.

To verify these responses in the continuous approach, we also compared the verum and sham response for all stimulations (verum) vs triggers (sham). We can confirm a clear amplitude enhancement when tone stimulations occurred compared to sham triggers (Figure 2A), again this response is more pronounced in strong than weak responders (Figure 2B). Finally, we wanted to see whether the ERPs differed between the windowed and continuous approach. Figure 2C illustrates that ERP of the windowed approach is more pronounced than the continuous approach during the verum nights, but not so during sham nights (Figure 2D) underlining the more efficient SWA enhancement in the windowed than continuous approach highlighted in manuscript Figure 4D.

Overall, the ERP results nicely confirms our previous results and help to explain the dose-dependent effects and the more efficient SWA enhancement in the windowed compared to the continuous approach. We added these figures to the supplement (Figure S2) and extended the results section with the following paragraphs:

Results:

“These results were supported by auditory evoked responses (Figure S2A) indicating that the averaged time series across stimulations were significantly different between conditions and the extent of the differences were more pronounced with increasing stimuli number per window for the ON windows and the difference between ON and OFF response, but not for the OFF-window triggers only. Specifically, with more stimulations per window the response got more pronounced in the temporal domain illustrating that subsequent stimuli further enhanced slow wave amplitudes and therefore, indicating a cumulative effect and a dose-dependent enhancement of lowSWA.”

“This was further confirmed by the evoked response analysis (Figure S2B) illustrating that the averaged difference between ON and OFF showed a more pronounced response for strong than weak responders during verum but not during sham.”

“Furthermore, comparing the evoked responses including all stimulations, the averaged time series significantly differed between verum and sham (Figure S4A) and this difference was more pronounced in strong than weak responders (Figure S4B).”

“This was further underlined by the analysis of the evoked responses indicating that the time series for the windowed approach showed significantly higher amplitudes than the continuous approach when comparing the verum condition (Figure S4C) but not when comparing sham (Figure S4D).”

Discussion:

“The evoked response analysis further underlined that more stimuli produced a more cumulative effect in time such that within a window more amplitude enhancements were found because of more subsequent stimuli.”

Methods:

“Auditory evoked responses of all stimuli (tones or triggers) were calculated. For the windowed approach these responses were analysed for all ON and OFF windows and again differentiated the windows for their respective number of stimulations. We also calculated the ON-OFF difference by subtracting the average OFF ERP from the ON ERP for the respective stimulation numbers.”

“For the evoked response analysis in supplementary Figures S2 and S4 we implemented a linear mixed-effect model with fixed factors condition (2 levels: verum, sham), night (7 levels), period order (2 levels), and nested random factor of subject and condition were included for Figures S2A and S4A. In Figures S2B and S4B we focused on the interaction between responder and condition. Figure S4C and D included the fixed factors approach (2 levels: ON, ONOFF), night (7 levels), period order (2 levels), and nested random factor of subject and approach. Please note that non-robust statistics were used here due to too extensive computation time for this analysis however, the p-values obtained were extremely small that independent of whether a robust or non-robust approach is used the outcomes would have been similar.”

Figure 1: Evoked responses (ERP) for verum and sham nights for the windowed approach with statistical results of linear mixed-effect models. (A) ERPs between sham and verum nights separately for ON (1st column), OFF (2nd column) and ON-OFF difference (3rd column), all for different stimulation numbers (row 1-3). P-values refer to the outputs of linear mixed-effect models for the factor condition (verum vs. sham). (B) ERP for all stimuli (tones or triggers) across conditions separately for weak and strong responders. P-values from linear mixed-effect model refer to the interaction condition x responder.

Figure 2: Evoked responses (ERP) for verum and sham nights with statistical results of linear mixed-effect models. (A) ERPs between sham and verum nights for the continuous approach including all stimulations (verum) or triggers (sham). P-values refer to the outputs of linear mixed effects models for the factor condition (verum vs. sham). (B) ERP for all stimuli (tones or triggers) across conditions separately for weak and strong responders for the continuous approach. P-values from linear mixed-effect model refer to the interaction condition x responder. (C) and (D) illustrate ERPs for all stimulations between the windowed and continuous approach for verum (C) and sham (D). P-values from linear mixed-effect model refer to the factor approach (ON vs ONOFF).

I. 132, It suffices to say: “In other 131 words, ON and OFF windows were quantized by the number of stimulations occurring during the ON and OFF windows.” Introducing “bins” without defining them impairs comprehension. Why not use “Stim No.” or similar as in the statistical test?

We adapted the sentence accordingly: “In other words, ON and OFF windows were quantized by the number of stimulations occurring during the ON and OFF windows (#stim)”.

I. 135, throughout: Is it a journal requirement to give the repeat the factors after “p”? They are already given with the F-value.

We agree with the reviewer that the mentioned factors after the p-values are unnecessary if the F-values already have the describing factors. We therefore removed them from the p-values if F-values were mentioned before to enhance readability.

150, add that these post-hoc analyses were exploratory since the superordinate ANOVA did not indicate a significant interaction.

We implemented the justified suggestion accordingly.

Fig. 2B, middle, Since the interaction cond x stim is not significant, the asterisks should at most be gray to indicate this. It is misleading to treat significant post hoc tests without an ANOVA effect the

same as those revealing a significant effect in the variance analyses. Also, I suggest using two asterisks for 3-4 stim, and only one for >4 stim in agreement with the p-values ($p < 0.01$ and $p < 0.05$).

We thank the reviewer for their helpful suggestion and used different asterix color and number to highlight the different p-value levels and the exploratory analysis. The figure legend was adapted accordingly.

L.187, “Before stimulation onset, the algorithm detected a 10-min window with robust NREM sleep and no stimulations occurring in which we calculated lowSWA.” Since the criteria for the first stimulation were a 10-min window with robust NREM sleep there is no need to write: “no stimulations occurring” in the sentence.

This makes indeed sense; we adapted the sentence accordingly.

I. 190-192. Although the Figure S2 looks convincing, information on subjects should be accompanied by their sleep stage, i.e., whether in N2 or N3. In addition, age should be given in the same graph. Could it be that older subjects had stable NREM with less N3? (cp. Also I.339-I.340)

The reviewer brings up a very interesting point. First, we added the age to the figure. Second, we explored in more detail the question about the sleep stages. Importantly, we expect to have sleep stage differences based on the different levels of slow wave activity and the fact that slow waves define our scoring distinction of N2 and N3. Because each night had different distributions of sleep stages during baseline within and across subjects, we made the same Figure S2 using % of N3 NREM sleep (of all NREM epoch in baseline) for color coding instead of lowSWA (Figure3, below). As expected, we clearly see that also sleep stages help to distinguish weak from strong responders. Nevertheless, when including the % of N3 in baseline NREM sleep as a predictor in the LDA it distinguished nights into weak and strong responders with an accuracy of 78% (Specificity = 0.88, Sensitivity = 0.69), which is highly significant but less predictive than with lowSWA. Overall, the main distinction of N2 and N3 NREM sleep is the number of slow waves with an amplitude of $75\mu\text{V}$ and therefore we expect a clear overlap of the findings. We believe that lowSWA is a more sensitive marker because it represents a continuous and objective metric of sleep depth available throughout sleep, while distinguishing NREM sleep stage N2 and N3 are discreet classes with artificial thresholds. We expanded the LDA analysis plot in Figure 3B of the paper and added a paragraph in the results referring to our points above:

“We further explored whether the amount of N3 NREM sleep (expressed as percentage) in the baseline could serve as a predictor in the LDA. It distinguished nights into weak and strong responders with an accuracy of 78% (Specificity = 0.88, Sensitivity = 0.69), which was highly significant but less predictive than with lowSWA. Overall, the main distinction of N2 and N3 NREM sleep is the number of slow waves with an amplitude of at least $75\mu\text{V}$ and therefore we expect a clear overlap of the findings. We believe that lowSWA is a more sensitive marker because it represents a continuous and objective metric of sleep depth available throughout sleep, while distinguishing NREM sleep stage N2 and N3 are discreet classes with artificial thresholds.”

Furthermore, we added the average amount of N3 NREM sleep (in percentage) across the sleep period per person below the age in supplementary Figure S3.

Figure 3: Individual ($n_{\text{Subjects}} = 16$) ON-OFF difference of lowSWA for all windows with >2 stimulations sorted by averaged lowSWA during verum in the windowed approach. Strong responders refer to the upper 50% and weak responder to the lower 50% of participants. Nights are color coded by the amount of N3 NREM sleep (expressed in percentage) in the baseline activity. Baseline corresponds to the first 10 min of stable NREM sleep detected during which the device was not performing stimulations. Age and averaged percentage of NREM sleep stage N3 (%N3) per night are summarized below each participant label. S: sham V: verum.

I. 213-223, please make the difference between rm and pm in text, and Rm and Pm in Fig. 3D clear. What exactly does each express? More information in the legend would also be helpful.

The figure rm pm are not correct, but the text one is. We apologize for this mistake. There is no significance in predicting the within subject differences with lowSWA, the way we described it in the text. We corrected the error in the figure accordingly. In addition, we started to consistently use a lower case r_{m} and p_{m} to avoid confusion due to inconsistencies in terminology.

I. 248 & fig. 4c: the inset of Fig. 4C should read “ >0.1 ” (as in the text), not “ >0.01 ”.

We thank the reviewer for this thorough reading and spotting this mistake. We changed it accordingly.

I. 250-251 “... lowSWA was on average increased in the windowed approach, specifically in strong responders.” This statement is an overemphasis, as only the interaction, but not post hoc tests reached significance.

We further clarified that in the text:

“Contrary to our hypothesis, lowSWE was on average increased in the windowed approach, specifically in strong responders, yet this was not significant.”

I.255-256, Fig. 4D, it would here also be important to see during which NREM sleep stage stimulation occurred, as sleep stage itself may affect responsiveness (e.g., Genzel, doi: 10.1016/j.tins.2013.10.002).

We thank the reviewer for their suggestion. To do so, we calculated the proportion of N3 NREM sleep epochs (in %) were included in the analysis of Figure 4D and compared this percentage between the different stimulation approaches for the verum condition in the strong responders. There was no significant difference ($F_{\text{appr}}(1,4,4) = 0.13, p > 0.1$) for verum ON (mean % of stimulated windows with N3 NREM sleep: $49.34\% \pm 4.19\%$) and verum ONOFF (mean % of stimulated windows with N3 NREM sleep: $48.17\% \pm 4.42\%$) in a robust linear mixed effects model analysis (fixed factors approach, nights, period-order, nested random factor of subject and approach).

I. 278-279 & Fig. 5, I assume decreased mood score corresponds to a more negative mood, but this should be explicitly stated in the results or figure legend.

We have added this missing information in the text and the Figure 5 legend.

I.291-293, “Finally, REM sleep changes were not predicted by changes in lowSWA during windows with more than 2 stimulations ($r_{rm} (179) = -0.06$, $p_{rm} > 0.1$, $CI = [-0.21; 0.09]$), further indicating that the effects of stimulation on slow waves are independent from the REM sleep and mood changes.” Based on one analysis, it seems a bit far-fetched to conclude that the change in lowSWA had no impact on REM sleep. Limit the result to no correlation between these two measures (cp. also 412-413).

We changed predicted to correlated in this sentence and toned down the conclusion as follows:

“Finally, REM sleep changes were not correlated to changes in lowSWA during windows with more than 2 stimulations ($r_{rm} (179) = -0.06$, $p_{rm} > 0.1$, $CI = [-0.21; 0.09]$), potentially indicating that the effects of stimulation on slow waves might be independent from the REM sleep and mood changes.”

“However, whether REM sleep causally influences mood needs further investigation. Importantly, neither REM changes nor mood changes were significantly correlated to lowSWA, the primary effect of auditory stimulation during sleep. Thus, secondary effects can occur that might be independent of the enhancement of slow waves and such possibilities need to be considered in the planning of future studies.”

I.314-315, for reasons given above it is misleading to conclude that “windows with more stimulations resulted in more pronounced enhancement of lowSWA on the group-level.” As it implies an increasing number of stimuli are beneficial. The main finding lies in the comparison of 1-2 vs. >2.

We thank the reviewer for this point and would like to clarify this statement. It is based on the strong findings in Figure 2A where we see that the difference between ON-OFF for windows increases with the increasing number of stimuli per window and that there is the strong condition and number of stimulation interaction for this figure. Based on the great suggestion of the reviewer we also saw this dose-dependency on the evoked response level. However, we agree that we might not have the resolution to justify that this increase is linear and the more stimuli the better. We therefore toned down this statement as follows:

“Within-night analysis in the windowed approach further provided some indication of a dose-dependent effect illustrating that windows with more stimulations (e.g., 1-2 stimuli vs >2 stimuli) resulted in more pronounced enhancement of lowSWA on the group-level.”

I. 500-501 were signals recorded as DC, or was there a hardware high pass filter? Which frequency was any low pass filter at recording?

We added the following, more detailed description of hardware/digital filtering while recording and saving the data in the methods paragraph:

“Biosignals were sampled at 250 Hz using a 24-bit analog-to-digital converter that featured an on-chip anti-aliasing filter. The raw biosignals including AC and DC components were saved to an SD card and further piped to the embedded real-time algorithms for auditory stimulation.”

I.507-508 please specify the direction of spectral power used to define NREM sleep. I imagine that both high OR low delta are high, and beta low? Give the frequency ranges here for all three bands.

Exactly, both delta values had to go above a threshold and the beta value below a threshold to be considered NREM sleep. We extended the method section with the following paragraph:

“First, the NREM sleep algorithm categorized the EEG data as NREM sleep or not-NREM sleep (awake and REM) based on the spectral power at different frequency bands: low delta ([2 - 4] Hz), high delta ([3 - 5] Hz), and high beta ([20 - 30] Hz). If low and high delta powers were above, and beta power below predefined and fixed thresholds during a specific window length, the EEG signal was classified as NREM sleep.”

I. 515 Are the 3 measures of sleep classification of performance metric given in percent? Please define these metrics.

The three measures refer to precision, specificity, and recall. Precision indicates the rate of correctly predicted samples as NREM sleep. Specificity indicates the ability of the sleep classification algorithms to identify not-NREM sleep correctly. Finally, the recall metric indicates the correct NREM sleep classification rate when the data was also scored as NREM sleep. All metrics are defined as relative numbers and are defined from 0 to 1.

We added the following information to Table supplementary Table 2:

“Precision, specificity, and recall are provided as relative values ranging from 0 to 1.”

And the following paragraph to the methods:

“Algorithm performance was quantified using precision, specificity and recall. Precision indicated the rate of correctly predicted NREM sleep samples. Specificity indicated the ability of the sleep classification algorithms to identify not-NREM sleep correctly. Finally, the recall metric indicated the correct NREM sleep classification rate when the data was also scored as NREM sleep. All metrics were defined as relative numbers ranging from 0 to 1.”

I.522, as different definitions of phase are used in the field, a figure of a slow wave with phases given is necessary. Supply the information whether 0° corresponds to the maximal down-state value or to the turning point from down-to-up state?

We thank the reviewer for this helpful suggestion and added the illustration below as an inset of Figure 1B.

I. 524-525, how long is the time window used to estimate EEG phase? Does preprocessing (before application of the algorithm) consist of any procedure other than 0.1 Hz high-pass filtering? - This is especially relevant if the manuscript of Ferster et al will not be published as open access.

The PLL algorithm is a control system that estimates the EEG phase in real time at each sample point. Therefore, there is no time window used for phase estimation. Since high-amplitude, low-frequency slow waves predominate during NREM sleep, the PLL can lock in phase with the EEG signal even in presence of other frequency components. Therefore, the PLL was applied in real time over the recorded and notch-filtered data that was further pre-processed with the high-pass filter at 0.1 Hz, avoiding extra undesired non-linear phase shifts induced by pre-processing filters.

We added the paragraph above to the manuscript but removed the reference to the Ferster et al. in preparation reference because the paper is under review but will likely not be published in due time.

I. 508, Were the predefined threshold values individualized? Were fixed quotients, e.g., of low delta to beta used?

The predefined thresholds were fixed and the same for all involved participants. We added the term fixed to the following paragraph:

“First, the NREM sleep algorithm categorized the EEG data as NREM sleep or not-NREM sleep (awake and REM) based on the spectral power at different frequency bands: low delta ([2 - 4] Hz), high delta ([3 - 5] Hz), and high beta ([20 - 30] Hz). If low and high delta powers were above, and beta power below predefined and fixed thresholds during a specific window length, the EEG signal was classified as NREM sleep.”

I. 543, Fig.1, What is the rationale behind using 6 sec for the windowed approach?

We aimed at a constant window length rather than a constant stimuli number to avoid different lengths of windows to compare within nights. We defined a length of 6s to be comparable to previous publications using an ON-OFF windowed approach with windows that had on average approximately the duration of 6s (e.g., Papalambros et al., 2017, Front. Hum. Neurosci, Papalambros et al., 2019, Ann. Clin. Transl. Neurol.).

We added the following to the methods section:

“We selected this window length to be comparable to previous publications using an ON-OFF windowed approach with windows that had on average the duration of $\sim 6 s^{24,36}$ ”

Papalambros, N. A. et al. Acoustic enhancement of sleep slow oscillations and concomitant memory improvement in older adults. Front. Hum. Neurosci. 11, 1–14 (2017).

Papalambros, N. A. et al. Acoustic enhancement of sleep slow oscillations in mild cognitive impairment. Ann. Clin. Transl. Neurol. 6, 1191–1201 (2019).

Fig. 1, “detected”? Isn’t “targeted” meant?

We replaced “detected” by “targeted” to increase consistency.

I.574, add that the values in Table S3 are per condition (as given in I. 608).

We have added this to the respective sentence.

Fig. 1, I.623, add a graph showing which EEG analyses periods are used for each of the 2 approaches, like Fig. 1B for recording. It is confusing, e.g., in I. 623 (analyses) to refer to: ON-OFF analyses AND consecutive sec analyses, since recording conditions are the same. Are 6 sec FFT epochs of the continuous approach meant with the latter?

We understand the confusion of the reviewer and realized that we were not detailed enough in explaining the used EEG analysis methods. Overall, there are two EEG analyses methods, both rely on the FFT analysis of 6-s windows. However, there is:

(1) ON-OFF analysis that can only be applied to the ONOFF condition (windowed stimulation condition). For this analysis approach, ON and OFF windows are selected based on the algorithm’s definition of ON and OFF and these windows are 6s in length. We used this analysis approach for Figures 2B, C, D and 3A, C, D

(2) Consecutive analysis can be applied to both the continuous and windowed stimulation approach. For this analysis the whole night is artificially divided in consecutive 6-s epochs and for each epoch the FFT is calculated and the number of stimuli within each epoch defined (thereof windows with e.g., >2 stimulations are selected). We used this approach for Figures 2A, 4-5. Of note, we also applied this analysis to the windowed stimulation approach to be a fair comparison to the continuous approach when we contrast these two. In addition, all spectral analysis of the robust NREM baseline (before stimulation starts) was using consecutive 6-s epochs.

We extended our method section extensively (see methods paragraph EEG analysis) to make a clear distinction of the analysis methods. Furthermore, we referred in every figure and text paragraph, which analysis method was employed to avoid any confusion.

Fig.2B, C Similar to the above, additional color/line style coding would be nice to assist in distinguishing comparisons of ON-OFF periods in the windowed stimulation condition (Fig. 2B, left, Fig. 2C) from the comparison ON- (OFF-) window period of stimulation and sham of the windowed approach Fig. 2B (middle, right).

We changed the line style suggested by the reviewers in Figure 2 and introduced a shaded version of the OFF in Figure 1 to increase readability and consistency across figures.

I.681, “was influenced” by ... is missing.

We thank the reviewer for catching this error and corrected it in the new version.

Inconsistent use of terms: I.385: “lowSWA power (lowSWE)”, yet power is depicted in figures aside from Fig. 4C.

We agree, the term power was removed from the sentence.

Three references are duplicated: 24 & 35, 25 & 37, 26 & 57.

Good catch, we thank the reviewer very much for finding these duplicates and we removed them.

Reviewer #2 (Remarks to the Author):

This is a well written, very practical report evaluating the effects of multi-night delivery of phase locked acoustic stimulation during deep sleep. The particular strengths are the multi-night data and the very careful analyses of factors determining inter-individual differences in responses to such stimulation : number of stimuli per window, total number of stimulations, baseline lowSWA and and the effects of mood.

I liked the analyses comparing whether stimulation effects were due to ON or OFF windows and the addition of the analysis of mood change.

There isn't much to comment on this polished but disappointing set of results other than to point out the average regression/ multi night correlation dashed line going in the opposite gradient direction as the rest of the data.

The authors are to be commended on a fine contribution.

We thank the reviewer very much for the positive feedback, the compliments, and the designated time to thoroughly read our manuscript.

We performed additional analysis and recognized in the course of re-analysing that there was a non-relevant miscalculation of the baseline window time. This very mildly affected some of the stimulation results. After correction we did not see any major changes that affected our presented results and conclusions. The changes in the text resulting from the re-calculation are further highlighted in green, all affected figures have been adapted.

We also wanted to provide further comments on the reviewer's observation regarding the repeated measures correlation in which the averaged group level dashed line can go in the opposite than the subject-level associations. This is for example observed in Figures 3D upper and middle figure. Importantly, repeated measures correlation only considers the intra-individual associations and the provided r - and p -value refers to the individual fitted lines and not on the dashed lines that represents the inter-individual association or group-level association. This analysis also highlights that it is important to distinguish inter- and intra-individual associations as they do not necessarily need to go in the same direction (see also Bakdash & Marusich, 2017). Using group-level statistics only (e.g., averaged correlations across subjects) intra-individual associations are not depicted. We also provided more information to the reader regarding this analysis in the methods section:

“Importantly, repeated measures correlation only considers the intra-individual associations and the provided r - and p -value refers to the individual fitted, solid lines in the presented figures and not on the dashed lines that represents the inter-individual association or group-level association. This analysis also highlights that it is important to distinguish inter- and intra-individual associations as they do not necessarily need to go in the same direction (see also Bakdash & Marusich, 2017).”

Bakdash, J. Z. & Marusich, L. R. Repeated measures correlation. *Front. Psychol.* 8, 456 (2017).

Reviewer #3 (Remarks to the Author):

In this manuscript, the authors have examined the auditory stimulation for elderly at home. The novelty of this work lies in the stimulation being applied at home using their portable monitoring/feedback-controlled modulation device. The authors have examined the effect of stimulation in sham and verum conditions. They found an enhancement of slow-wave activity after stimulation. However, there was an adverse effect on the duration of REM sleep and subjective mood. Overall, the manuscript is scientifically sound and is very well-written.

It is unfortunate that the number of participants had to be limited to 16 due to Covid. Nonetheless, the effects appear to be robust even with a small sample size.

Typo on line 359 - yAlzheimer's -> Alzheimer's.

We thank the reviewer very much for the positive feedback, the compliments, and the designated time to thoroughly read our manuscript. We corrected the mentioned typo accordingly. We added the limitation of the small sample size to the discussion section by extending the following sentence:

“Even though we had a small sample size due to COVID-19 related reasons, this finding is specifically robust because we reproduced it in two different stimulation approaches, windowed and continuous, in the same individuals over multiple consecutive nights.”

We performed additional analysis and recognized in the course of re-analysing that there was a non-relevant miscalculation of the baseline window time. This very mildly affected some of the stimulation results. After correction we did not see any major changes that affected our presented results and conclusions. The changes in the text resulting from the re-calculation are further highlighted in green, all affected figures have been adapted.

Reviewers' comments:

Reviewer #1 (Remarks to the Author):

All my concerns were addressed. Very nice.
Only, shouldn't it be, on New line 135, "(stim#)"?

Reviewer #2 (Remarks to the Author):

The authors answered my question about the within subject and overall (between -subject) multi-correlation between baseline SWA and ON-OFF lowSWA difference. However, this brings attention to a difficulty in understanding what the key message regarding stimulation is. In fig 2B and 3C in the Verum condition, more stimulation per window relates to larger lowSWA ON-OFF difference. Yet in Fig, 3D, the discussion and the response to reviewer #1, they say more stimulations is not necessarily better. The only way I can reconcile these two findings is that the stronger responders have fewer slow waves (that can be stimulated) but which, when stimulated, achieve higher changes in amplitude. If this is so, can the authors say it. Otherwise a simplifying explanation in the text would be useful

On re-examining, the comparison between windowed and continuous approaches to stimulation while . I believe the author's surmise that windowed is superior but their data really doesn't have the power to show this.

For the non-specialist reader, the main message of this work beyond the technical details is that acoustic stimulation can work but only in younger people, who are not the ones who benefit most from augmenting slow waves. This seems in line with the ideas expressed in the reviewer of Wunderlun; Sleep 2021 but with lots more detail and justification. The authors might consider adding a summary statement to this effect where the reader is likely to see it.

The inclusion of the ERP data in the supplementary material is brilliant but reveals the very slight change in SW as a result of stimulation (no surprises for those we have used this method). Along this line, with the small sample size of 16, perhaps the labels should be shifted to 'responder' and non-responder'. As there is no behavioural benefit to even in the 'responders' here, (PVT, daytime sleepiness, sleep quality). This would be appropriate.

The legend in Fig 4B is either wrong- the comparison is between windowed and continuous approaches or the authors have inserted the wrong bar graph (which refers to sham vs. verum)

Reviewers' comments: Point-by-point reply

Reviewer #1 (Remarks to the Author):

All my concerns were addressed. Very nice.

Only, shouldn't it be, on New line 135, "(stim#)"?

We thank the reviewer very much for the positive feedback, the compliments, and the designated time to thoroughly read our manuscript.

We have changed the new line 135 from (#stim) to (stim#).

Reviewer #2 (Remarks to the Author):

The authors answered my question about the within subject and overall (between -subject) multi-correlation between baseline SWA and ON-OFF lowSWA difference.

We thank the reviewer for their valuable and thorough feedback, and we highly appreciated the constructive inputs. Please find our statements to the individual points below:

However, this brings attention to a difficulty in understanding what the key message regarding stimulation is. In fig 2B and 3C in the Verum condition, more stimulation per window relates to larger lowSWA ON-OFF difference. Yet in Fig, 3D, the discussion and the response to reviewer #1, they say more stimulations is not necessarily better. The only way I can reconcile these two findings is that the stronger responders have fewer slow waves (that can be stimulated) but which, when stimulated, achieve higher changes in amplitude. If this is so, can the authors say it. Otherwise a simplifying explanation in the text would be useful

The reviewer brings up an important point. Indeed, these results seem to be conflicting initially. But there is a possible explanation that might relate to the timescale we look at. Figure 2B and 2C illustrate that with increasing number of stimuli per window there are more pronounced SWA changes supporting the idea that more stimulations are leading to more pronounced responses. This is also represented in the ERP analysis because looking at a short time window of a few seconds (e.g., ON or OFF windows) we can clearly depict that amplitude enhancements of more temporal consecutive waves are observed compared to windows with less stimuli. However, if we compare the windowed to the continuous approach in which we have the timescale of a night rather than a few seconds, we see no difference of SWE in all stimulated windows even though we have double the number of stimulations for the continuous approach. This is further confirmed by the findings that the SWA response between continuous and windowed approach in windows with more than 2 stimulations are more pronounced in the windowed approach and also the ERP is slightly but significantly more pronounced in the windowed approach compared to the continuous one. This points to the idea that more might not necessarily be better if no defined breaks are introduced as in the windowed approach. Thus, in a window of a few seconds more stimuli might be beneficial, however more continued stimulation without defined breaks could possibly lead to an adaptation to the stimulation and responsiveness to tones is reduced.

We extended the following paragraph in the discussion (red marked):

“Indeed, in strong responders lowSWA was more pronounced during verum in the windowed approach than in the continuous approach. Our results therefore point to more effective enhancement of SWA in the windowed approach if we only consider short-term timeframes, such as the 6 s windows. Of note,

averaged epochs do not provide any information regarding overall enhancement of lowSWA across the stimulation period. Investigating the cumulated lowSWA (lowSWE) over the whole detected NREM period, there was no significant difference between the continuous and windowed approach emphasizing that at this stage we cannot term one of the approaches as superior. Interestingly, this finding seemingly contradicts the idea that there is a dose-dependent improvement, because lowSWE is not more enhanced in the continuous approach despite having around double the number of stimulations compared to the windowed approach. Yet, we suppose that a dose-dependent effect might specifically be observable in a short timeframe of a few seconds but in the perspective of a whole night more stimulations might reduce responsiveness. Thus, it is likely a trade-off between finding the ideal amount of stimulations and obtaining sufficient breaks between stimulation trains to maximize effectiveness in enhancing lowSWA and lowSWE across nights. This should be systematically investigated in future studies.”

On re-examining, the comparison between windowed and continuous approaches to stimulation while . I believe the author's surmise that windowed is superior but their data really doesn't have the power to show this.

We thank the reviewer for raising this point. We agree that at this stage we cannot term any of the used approaches as superior because there is no difference in SWE across the stimulation period between the conditions. In this regard, our study might have been underpowered. However, even though around double the number of stimulations were applied in the continuous approach compared to the windowed, the continuous was not more effective. Why is this the case? In a more short-term perspective (e.g., windows of a few seconds) we have clear statistical evidence that the windowed approach is more effective because lowSWA was more pronouncedly enhanced in windows with >2 stimulations in the windowed compared to the continuous approach. Furthermore, the ERP response was slightly but significantly more pronounced in the windowed than the continuous approach, again pointing to more effective enhancement with the windowed approach.

We emphasized this for the reader by modifying the following paragraph in the discussion (red marked):

“Indeed, in strong responders lowSWA was more pronounced during verum in the windowed approach than in the continuous approach. Our results therefore point to more effective enhancement of SWA in the windowed approach if we only consider short-term timeframes, such as the 6 s windows. Of note, averaged epochs do not provide any information regarding overall enhancement of lowSWA across the stimulation period. Investigating the cumulated lowSWA (lowSWE) over the whole detected NREM period, there was no significant difference between the continuous and windowed approach emphasizing that at this stage we cannot term one of the approaches as superior. Interestingly, this finding seemingly contradicts the idea that there is a dose-dependent improvement, because lowSWE is not more enhanced in the continuous approach despite having around double the number of stimulations compared to the windowed approach. Yet, we suppose that a dose-dependent effect might specifically be observable in a short timeframe of a few seconds but in the perspective of a whole night more stimulations might reduce responsiveness. Thus, it is likely a trade-off between finding the ideal amount of stimulations and obtaining sufficient breaks between stimulation trains to maximize effectiveness in enhancing lowSWA and lowSWE across nights. This should be systematically investigated in future studies.”

For the non-specialist reader, the main message of this work beyond the technical details is that acoustic stimulation can work but only in younger people, who are not the ones who benefit most from augmenting slow waves. This seems in line with the ideas expressed in the reviewer of Wunderlun; Sleep 2021 but with lots more detail and justification. The authors might consider adding a summary statement to this effect where the reader is likely to see it. We understand the point raised by the reviewer, but we do not think that our dataset is optimized to provide conclusive statements regarding age differences. First, all participants were in the older age range that is over 60 years of age and the age difference between strong and weak responders is small (a few years). We would therefore not necessarily term any of them as young people. Second, we believe that not age but rather the remaining SWA level defines how much we can augment SWA. Of course,

slow waves change as a function of age with a decrease of SWA with higher age. Therefore, even in our dataset that does not span a wide age range (62-78 years of age) we see differences in age that are mirrored in SWA differences. Nevertheless, looking closely at Figure S3, we see for instance S030 that is age wise rather belonging to the “younger group” and in the age range of the strong responders. However, this participant has a very low level of baseline SWA and consequently also no clear response to auditory stimuli. We agree with the reviewer that from a clinical perspective, we would also like to modulate people that have specifically lowSWA but our results raise doubts of whether this can be effectively done using our stimulation approach in patients that fall below a certain SWA level. We already highlight that in our discussion. We further added the citation of Wunderlin et al. 2021 in our discussion and thank the reviewer for providing this insightful systematic review and meta-analysis.

The inclusion of the ERP data in the supplementary material is brilliant but reveals the very slight change in SW as a result of stimulation (no surprises for those we have used this method). Along this line, with the small sample size of 16, perhaps the labels should be shifted to 'responder' and non-responder'. As there is no behavioural benefit to even in the 'responders' here, (PVT, daytime sleepiness, sleep quality). This would be appropriate. We thank the reviewer for this compliment and agree that the ERP analysis that was suggested by Reviewer 1 provided many interesting insights. Indeed, the amplitude changes are small but highly significantly different and, as the reviewer points out, in the expected range. We also have to consider that averaging in time over multiple EEG traces with different shapes, the effects might be underestimated as some effects might average out. We therefore believe that it is crucial to complement ERP analysis with SWA information because it is not affected by temporal averaging. We acknowledge the idea of the reviewer to rename our groups and in an initial version of our manuscript we also used this terminology of responder vs. non-responder. However, we got feedback from other experts that this distinction is not ideal because (1) statistically it is very difficult to show no differences, and (2) also weak responders show a difference in ERP/SWA analysis between conditions, just strongly reduced in magnitude. We therefore believe it is more appropriate to use the already established terminology and hope the reviewer agrees with this. Of note, the weak vs strong responders were only differentiated based on the effect of auditory stimuli on SWA that is our primary outcome and are not including any information about behavioural benefit in their definition.

The legend in Fig 4B is either wrong- the comparison is between windowed and continuous approaches or the authors have inserted the wrong bar graph (which refers to sham vs. verum) We thank the reviewer for raising this point. However, Figure 4B and its legend are correct and do match the description in the manuscript. In Figure 4A and 4B, we directly compare the verum and sham for the continuous approach, reproducing the analysis for the windowed approach. Importantly, for the continuous approach there is no ON-OFF difference because we do not employ these OFF breaks in the continuous approach. We therefore have bar graphs that refer to defined windows with more than 2 stimulations (see methods for more details). Figures 4C and 4D illustrating the comparison between the continuous and windowed approaches. If the reviewer has a suggestion of what information could further help to make it clearer, we are happy to implement that.

However, we have realized that in Figure S4D the conditions are termed verum instead of sham (the figure caption is correct, however the legend next to the graph says verum). We corrected this accordingly.

REVIEWERS' COMMENTS:

Reviewer #2 (Remarks to the Author):

The authors have addressed my concerns.